# SPACE: Unsupervised Object-Oriented Scene Representation via Spatial Attention and Decomposition

{**Zhixuan Lin**[1,2*]**, Yi-Fu Wu**[1]**, Skand Vishwanath Peri**[1]**,}**
**Weihao Sun**[1]**, Gautam Singh**[1]**, Fei Deng**[1]**, Jindong Jiang**[1]**, Sungjin Ahn**[1]

[1]Rutgers University & [2]Zhejiang University

## Abstract

The ability to decompose complex multi-object scenes into meaningful abstractions like objects is fundamental to achieve higher-level cognition. Previous approaches for *unsupervised object-oriented scene representation learning* are either based on spatial-attention or scene-mixture approaches and limited in scalability which is a main obstacle towards modeling real-world scenes. In this paper, we propose a generative latent variable model, called SPACE, that provides a unified probabilistic modeling framework that combines the best of spatial-attention and scene-mixture approaches. SPACE can explicitly provide factorized object representations for foreground objects while also decomposing background segments of complex morphology. Previous models are good at either of these, but not both. SPACE also resolves the scalability problems of previous methods by incorporating parallel spatial-attention and thus is applicable to scenes with a large number of objects without performance degradations. We show through experiments on Atari and 3D-Rooms that SPACE achieves the above properties consistently in comparison to SPAIR, IODINE, and GENESIS. Results of our experiments can be found on our project website: https://sites.google.com/view/space-project-page

## 1 Introduction

One of the unsolved key challenges in machine learning is unsupervised learning of structured representation for a visual scene containing many objects with occlusion, partial observability, and complex background. When properly decomposed into meaningful abstract entities such as objects and spaces, this structured representation brings many advantages of abstract (symbolic) representation to areas where contemporary deep learning approaches with a global continuous vector representation of a scene have not been successful. For example, a structured representation may improve sample efficiency for downstream tasks such as a deep reinforcement learning agent (Mnih et al., 2013). It may also enable visual variable binding (Sun, 1992) for reasoning and causal inference over the relationships between the objects and agents in a scene. Structured representations also provide composability and transferability for better generalization.

Recent approaches to this problem of *unsupervised object-oriented scene representation* can be categorized into two types of models: *scene-mixture* models and *spatial-attention* models. In scene-mixture models (Greff et al., 2017; 2019; Burgess et al., 2019; Engelcke et al., 2019), a visual scene is explained by a mixture of a finite number of component images. This type of representation provides flexible segmentation maps that can handle objects and background segments of complex morphology. However, since each component corresponds to a full-scale image, important physical features of objects like position and scale are only implicitly encoded in the scale of a full image and further disentanglement is required to extract these useful features. Also, since it does not explicitly reflect useful inductive biases like the locality of an object in the Gestalt principles (Koffka, 2013),

---

*Visiting Student at Rutgers University. Authors named inside {} equally contributed. Correspondance to zxlin.cs@gmail.com and sjn.ahn@gmail.com.

the resulting component representation is not necessarily a representation of a local area. Moreover, to obtain a complete scene, a component needs to refer to other components, and thus inference is inherently performed sequentially, resulting in limitations in scaling to scenes with many objects.

In contrast, spatial-attention models (Eslami et al., 2016; Crawford & Pineau, 2019) can explicitly obtain the fully disentangled geometric representation of objects such as position and scale. Such features are grounded on the semantics of physics and should be useful in many ways (e.g., sample efficiency, interpretability, geometric reasoning and inference, transferability). However, these models cannot represent complex objects and background segments that have too flexible morphology to be captured by spatial attention (i.e. based on rectangular bounding boxes). Similar to scene-mixture models, previous models in this class show scalability issues as objects are processed sequentially.

In this paper, we propose a method, called *Spatially Parallel Attention and Component Extraction* (SPACE), that combines the best of both approaches. SPACE learns to process foreground objects, which can be captured efficiently by bounding boxes, by using parallel spatial-attention while decomposing the remaining area that includes both morphologically complex objects and background segments by using component mixtures. Thus, SPACE provides an object-wise disentangled representation of foreground objects along with explicit properties like position and scale per object while also providing decomposed representations of complex background components. Furthermore, by fully parallelizing the foreground object processing, we resolve the scalability issue of existing spatial attention methods. In experiments on 3D-room scenes and Atari game scenes, we quantitatively and qualitatively compare the representation of SPACE to other models and show that SPACE combines the benefits of both approaches in addition to significant speed-ups due to the parallel foreground processing.

The contributions of the paper are as follows. First, we introduce a model that unifies the benefits of spatial-attention and scene-mixture approaches in a principled framework of probabilistic latent variable modeling. Second, we introduce a spatially parallel multi-object processing module and demonstrate that it can significantly mitigate the scalability problems of previous methods. Lastly, we provide an extensive comparison with previous models where we illustrate the capabilities and limitations of each method.

## 2   THE PROPOSED MODEL: SPACE

In this section, we describe our proposed model, Spatially Parallel Attention and Component Extraction (SPACE). The main idea of SPACE, presented in Figure 1, is to propose a unified probabilistic generative model that combines the benefits of the spatial-attention and scene-mixture models.

### 2.1   GENERATIVE PROCESS

SPACE assumes that a scene $\mathbf{x}$ is decomposed into two independent latents: foreground $\mathbf{z}^{\text{fg}}$ and background $\mathbf{z}^{\text{bg}}$. The foreground is further decomposed into a set of independent foreground objects $\mathbf{z}^{\text{fg}} = \{\mathbf{z}_i^{\text{fg}}\}$ and the background is also decomposed further into a sequence of background segments $\mathbf{z}^{\text{bg}} = \mathbf{z}_{1:K}^{\text{bg}}$. While our choice of modeling the foreground and background independently worked well empirically, for better generation, it may also be possible to condition one on the other. The image distributions of the foreground objects and the background components are combined together with a pixel-wise mixture model to produce the complete image distribution:

$$p(\mathbf{x}|\mathbf{z}^{\text{fg}}, \mathbf{z}^{\text{bg}}) = \alpha \underbrace{p(\mathbf{x}|\mathbf{z}^{\text{fg}})}_{\text{Foreground}} + (1 - \alpha) \sum_{k=1}^{K} \pi_k \underbrace{p(\mathbf{x}|\mathbf{z}_k^{\text{bg}})}_{\text{Background}}. \tag{1}$$

Here, the foreground mixing probability $\alpha$ is computed as $\alpha = f_\alpha(\mathbf{z}^{\text{fg}})$. This way, the foreground is given precedence in assigning its own mixing weight and the remaining is apportioned to the background. The mixing weight assigned to the background is further sub-divided among the $K$ background components. These weights are computed as $\pi_k = f_{\pi_k}(\mathbf{z}_{1:k}^{\text{bg}})$ and $\sum_k \pi_k = 1$. With these notations, the complete generative model can be described as follows.

$$p(\mathbf{x}) = \iint p(\mathbf{x}|\mathbf{z}^{\text{fg}}, \mathbf{z}^{\text{bg}})p(\mathbf{z}^{\text{bg}})p(\mathbf{z}^{\text{fg}})d\mathbf{z}^{\text{fg}}d\mathbf{z}^{\text{bg}} \tag{2}$$

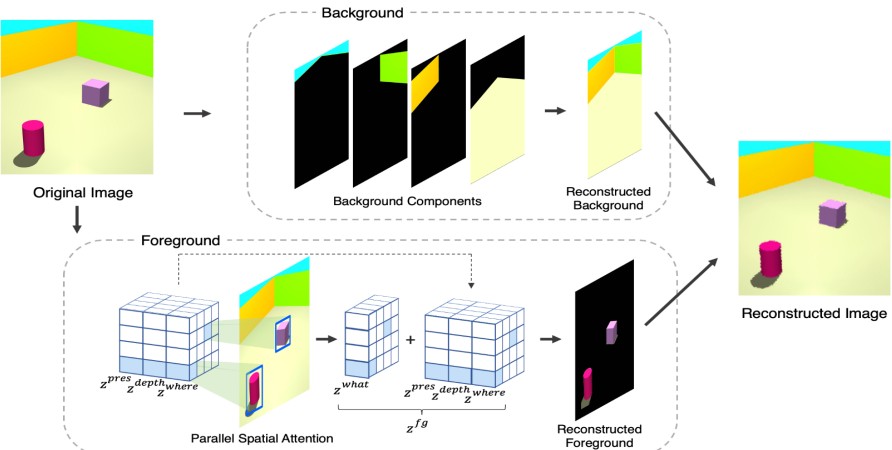

**Figure 1:** Illustration of the SPACE model. SPACE consists of a foreground module and a background module. In the foreground module, the input image is divided into a grid of $H \times W$ cells ($4 \times 4$ in the figure). An image encoder is used to compute the $z^{\text{where}}$, $z^{\text{depth}}$, and $z^{\text{pres}}$ for each cell in parallel. $z^{\text{where}}$ is used to identify proposal bounding boxes and a spatial transformer is used to attend to each bounding box in parallel, computing a $z^{\text{what}}$ encoding for each cell. The model selects patches using the bounding boxes and reconstructs them using a VAE from all the foreground latents $z^{\text{fg}}$. The background module segments the scene into $K$ components (4 in the figure) using a pixel-wise mixture model. Each component consists of a set of latents $z^{\text{bg}} = (z^m, z^c)$ where $z^m$ models the mixing probability of the component and $z^c$ models the RGB distribution of the component. The components are combined to reconstruct the background using a VAE. The reconstructed background and foreground are then combined using a pixel-wise mixture model to generate the full reconstructed image.

We now describe the foreground and background models in more detail.

**Foreground.** SPACE implements $\mathbf{z}^{\text{fg}}$ as a structured latent. In this structure, an image is treated as if it were divided into $H \times W$ cells and each cell is tasked with modeling at most one (nearby) object in the scene. This type of structuring has been used in (Redmon et al., 2016; Santoro et al., 2017; Crawford & Pineau, 2019). Similar to SPAIR, in order to model an object, each cell $i$ is associated with a set of latents $(\mathbf{z}_i^{\text{pres}}, \mathbf{z}_i^{\text{where}}, \mathbf{z}_i^{\text{depth}}, \mathbf{z}_i^{\text{what}})$. In this notation, $\mathbf{z}^{\text{pres}}$ is a binary random variable denoting if the cell models any object or not, $\mathbf{z}^{\text{where}}$ denotes the size of the object and its location relative to the cell, $\mathbf{z}^{\text{depth}}$ denotes the *depth* of the object to resolve occlusions and $\mathbf{z}^{\text{what}}$ models the object appearance and its mask. These latents may then be used to compute the foreground image component $p(\mathbf{x}|\mathbf{z}^{\text{fg}})$ which is modeled as a Gaussian distribution $\mathcal{N}(\mu^{\text{fg}}, \sigma_{\text{fg}}^2)$. In practice, we treat $\sigma_{\text{fg}}^2$ as a hyperparameter and decode only the mean image $\mu^{\text{fg}}$. In this process, SPACE reconstructs the objects associated to each cell having $\mathbf{z}_i^{\text{pres}} = 1$. For each such cell, the model uses the $\mathbf{z}_i^{\text{what}}$ to decode the object glimpse and its mask and the glimpse is then positioned on a full-resolution canvas using $\mathbf{z}_i^{\text{where}}$ via the Spatial Transformer (Jaderberg et al., 2015). Using the object masks and $\mathbf{z}_i^{\text{depth}}$, all the foreground objects are combined into a single foreground mean-image $\mu^{\text{fg}}$ and the foreground mask $\alpha$ (See Appendix D for more details).

SPACE imposes a prior distribution on these latents as follows:

$$p(\mathbf{z}^{\text{fg}}) = \prod_{i=1}^{H \times W} p(\mathbf{z}_i^{\text{pres}}) \left( p(\mathbf{z}_i^{\text{where}}) p(\mathbf{z}_i^{\text{depth}}) p(\mathbf{z}_i^{\text{what}}) \right)^{\mathbf{z}_i^{\text{pres}}} \tag{3}$$

Here, only $\mathbf{z}_i^{\text{pres}}$ is modeled using a Bernoulli distribution while the remaining are modeled as Gaussian.

**Background.** To model the background, SPACE implements $\mathbf{z}_k^{\text{bg}}$, similar to GENESIS, as $(\mathbf{z}_k^m, \mathbf{z}_k^c)$ where $\mathbf{z}_k^m$ models the mixing probabilities $\pi_k$ of the components and $\mathbf{z}_k^c$ models the RGB distribution $p(\mathbf{x}|\mathbf{z}_k^{\text{bg}})$ of the $k^{\text{th}}$ background component as a Gaussian $\mathcal{N}(\mu_i^{\text{bg}}, \sigma_{\text{bg}}^2)$. The following prior is

imposed upon these latents.

$$p(\mathbf{z}^{\mathrm{bg}}) = \prod_{k=1}^{K} p(\mathbf{z}_k^c|\mathbf{z}_k^m)p(\mathbf{z}_k^m|\mathbf{z}_{<k}^m) \tag{4}$$

## 2.2 INFERENCE AND TRAINING

Since we cannot analytically evaluate the integrals in equation 2 due to the continuous latents $\mathbf{z}^{\mathrm{fg}}$ and $\mathbf{z}_{1:K}^{\mathrm{bg}}$, we train the model using a variational approximation. The true posterior on these variables is approximated as follows.

$$p(\mathbf{z}_{1:K}^{\mathrm{bg}}, \mathbf{z}^{\mathrm{fg}}|\mathbf{x}) \approx q(\mathbf{z}^{\mathrm{fg}}|\mathbf{x}) \prod_{k=1}^{K} q(\mathbf{z}_k^{\mathrm{bg}}|\mathbf{z}_{<k}^{\mathrm{bg}}, \mathbf{x}) \tag{5}$$

This is used to derive the following ELBO to train the model using the reparameterization trick and SGD (Kingma & Welling, 2013).

$$\mathcal{L}(\mathbf{x}) = \mathbb{E}_{q(\mathbf{z}^{\mathrm{fg}},\mathbf{z}^{\mathrm{bg}}|\mathbf{x})} \Big[ \log p(\mathbf{x}|\mathbf{z}^{\mathrm{fg}}, \mathbf{z}^{\mathrm{bg}}) - \sum_{k=1}^{K} D_{\mathrm{KL}}(q(\mathbf{z}_k^{\mathrm{bg}}|\mathbf{z}_{<k}^{\mathrm{bg}}, \mathbf{x}) \parallel p(\mathbf{z}_k^{\mathrm{bg}}|\mathbf{z}_{<k}^{\mathrm{bg}}))$$
$$- \sum_{i=1}^{H \times W} D_{\mathrm{KL}}(q(\mathbf{z}_i^{\mathrm{fg}}|\mathbf{x}) \parallel p(\mathbf{z}_i^{\mathrm{fg}})) \Big] \tag{6}$$

See Appendix B for the detailed decomposition of the ELBO and the related details.

**Parallel Inference of Cell Latents.** SPACE uses mean-field approximation when inferring the cell latents, so $\mathbf{z}_i^{\mathrm{fg}} = (\mathbf{z}_i^{\mathrm{pres}}, \mathbf{z}_i^{\mathrm{where}}, \mathbf{z}_i^{\mathrm{depth}}, \mathbf{z}_i^{\mathrm{what}})$ for each cell does not depend on other cells.

$$q(\mathbf{z}^{\mathrm{fg}}|\mathbf{x}) = \prod_{i=1}^{H \times W} q(\mathbf{z}_i^{\mathrm{pres}}|\mathbf{x}) \left( q(\mathbf{z}_i^{\mathrm{where}}|\mathbf{x})q(\mathbf{z}_i^{\mathrm{depth}}|\mathbf{x})q(\mathbf{z}_i^{\mathrm{what}}|\mathbf{z}_i^{\mathrm{where}}, \mathbf{x}) \right)^{\mathbf{z}_i^{\mathrm{pres}}} \tag{7}$$

As shown in Figure 1, this allows each cell to act as an independent object detector, spatially attending to its own local region in parallel. This is in contrast to inference in SPAIR, where each cell's latents auto-regressively depend on some or all of the previously traversed cells in a row-major order i.e., $q(\mathbf{z}^{\mathrm{fg}}|\mathbf{x}) = \prod_{i=1}^{HW} q(\mathbf{z}_i^{\mathrm{fg}}|\mathbf{z}_{<i}^{\mathrm{fg}}, \mathbf{x})$. However, this method becomes prohibitively expensive in practice as the number of objects increases. While Crawford & Pineau (2019) claim that these lateral connections are crucial for performance since they model dependencies between objects and thus prevent duplicate detections, we challenge this assertion by observing that 1) due to the bottom-up encoding conditioning on the input image, each cell should have information about its nearby area without explicitly communicating with other cells, and 2) in (physical) spatial space, two objects cannot exist at the same position. Thus, the relation and interference between objects should not be severe and the mean-field approximation is a good choice in our model. In our experiments, we verify empirically that this is indeed the case and observe that SPACE shows comparable detection performance to SPAIR while having significant gains in training speeds and efficiently scaling to scenes with many objects.

**Preventing Box-Splitting.** If the prior for the bounding box size is set to be too small, then the model could split a large object by multiple bounding boxes and when the size prior is too large, the model may not capture small objects in the scene, resulting in a trade-off between the prior values of the bounding box size. To alleviate this problem, we found it sometimes helpful to introduce an auxiliary loss which we call the *boundary loss*. In the boundary loss, we construct a boundary of thickness $b$ pixels along the borders of each glimpse. Then, we restrict an object to be inside this boundary and penalize the model if an object's mask overlaps with the boundary area. Thus, the model is penalized if it tries to split a large object by multiple smaller bounding boxes. A detailed implementation of the boundary loss is mentioned in Appendix C.

## 3 RELATED WORKS

Our proposed model is inspired by several recent works in unsupervised object-oriented scene decomposition. The Attend-Infer-Repeat (AIR) (Eslami et al., 2016) framework uses a recurrent neural network to attend to different objects in a scene and each object is sequentially processed one

at a time. An object-oriented latent representation is prescribed that consists of 'what', 'where', and 'presence' variables. The 'what' variable stores the appearance information of the object, the 'where' variable represents the location of the object in the image, and the 'presence' variable controls how many steps the recurrent network runs and acts as an interruption variable when the model decides that all objects have been processed.

Since the number of steps AIR runs scales with the number of objects it attends to, it does not scale well to images with many objects. Spatially Invariant Attend, Infer, Repeat (SPAIR) (Crawford & Pineau, 2019) attempts to address this issue by replacing the recurrent network with a convolutional network. Similar to YOLO (Redmon et al., 2016), the locations of objects are specified relative to local grid cells rather than the entire image, which allow for spatially invariant computations. In the encoder network, a convolutional neural network is first used to map the image to a feature volume with dimensions equal to a pre-specified grid size. Then, each cell of the grid is processed *sequentially* to produce objects. This is done sequentially because the processing of each cell takes as input feature vectors and sampled objects of nearby cells that have already been processed. SPAIR therefore scales with the pre-defined grid size which also represents the maximum number of objects that can be detected. Our model uses an approach similar to SPAIR to detect foreground objects, but importantly we make the foreground object processing fully parallel to scale to large number of objects without performance degradation. Works based on Neural Expectation Maximization (Van Steenkiste et al., 2018; Greff et al., 2017) do achieve unsupervised object detection but do not explicitly model the *presence*, *appearance*, and *location* of objects. These methods also suffer from the problem of scaling to images with a large number of objects. In a related line of research, AIR has also recently been extended to track objects in a sequence of images (Kosiorek et al., 2018; Crawford & Pineau, 2020; Jiang et al., 2020).

For unsupervised scene-mixture models, several recent models have shown promising results. MONet (Burgess et al., 2019) leverages a deterministic recurrent attention network that outputs pixel-wise masks for the scene components. A variational autoencoder (VAE) (Kingma & Welling, 2013) is then used to model each component. IODINE (Greff et al., 2019) approaches the problem from a spatial mixture model perspective and uses amortized iterative refinement of latent object representations within the variational framework. GENESIS (Engelcke et al., 2019) also uses a spatial mixture model which is encoded by component-wise latent variables. Relationships between these components are captured with an autoregressive prior, allowing complete images to be modeled by a collection of components.

## 4 EVALUATION

We evaluate our model on two datasets: 1) an Atari (Bellemare et al., 2013) dataset that consists of random images from a pretrained agent playing the games, and 2) a generated 3D-room dataset that consists of images of a walled enclosure with a random number of objects on the floor. In order to test the scalability of our model, we use both a small 3D-room dataset that has 4-8 objects and a large 3D-room dataset that has 18-24 objects. Each image is taken from a random camera angle and the colors of the objects, walls, floor, and sky are also chosen at random. Additional details of the datasets can be found in the Appendix E.

**Baselines.** We compare our model against two scene-mixture models (IODINE and GENESIS) and one spatial-attention model (SPAIR). Since SPAIR does not have an explicit background component, we add an additional VAE for processing the background. Additionally, we test against two implementations of SPAIR: one where we train on the entire image using a $16 \times 16$ grid and another where we train on random $32 \times 32$ pixel patches using a $4 \times 4$ grid. We denote the former model as SPAIR and the latter as SPAIR-P. SPAIR-P is consistent with the SPAIR's alternative training regime on Space Invaders demonstrated in Crawford & Pineau (2019) to address the slow training of SPAIR on the full grid size because of its sequential inference. Lastly, for performance reasons, unlike the original SPAIR implementation, we use parallel processing for rendering the objects from their respective latents onto the canvas[1] for both SPAIR and SPAIR-P. Thus, because of these improvements, our SPAIR implementation can be seen as a stronger baseline than the original SPAIR. More details of the baselines are given in Appendix D.

---

[1] It is important to note that the worst case complexity of rendering is $\mathcal{O}(hw \times HW)$, (where $(h, w)$ is the image size) which is extremely time consuming when we have large image size and/or large number of objects.

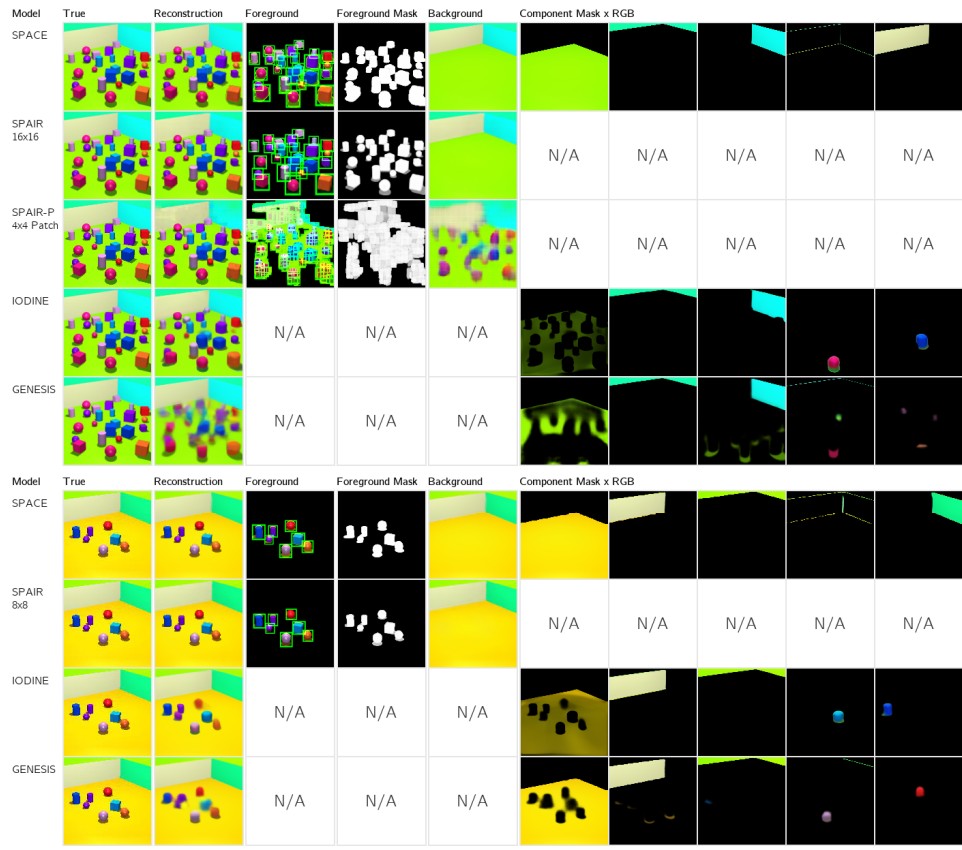

**Figure 2:** Qualitative comparison between SPACE , SPAIR, SPAIR-P, IODINE and GENESIS for the 3D-Room dataset.

## 4.1 QUALITATIVE COMPARISON OF INFERRED REPRESENTATIONS

In this section, we provide a qualitative analysis of the generated representations of the different models. For each model, we performed a hyperparameter search and present the results for the best settings of hyperparameters for each environment. Figure 2 shows sample scene decompositions of our baselines from the 3D-Room dataset and Figure 3 shows the results on Atari. Note that SPAIR does not use component masks and IODINE and GENESIS do not separate foreground from background, hence the corresponding cells are left empty. Additionally, we only show a few representative components for IODINE and GENESIS since we ran those experiments with larger $K$ than can be displayed. More qualitative results of SPACE can be found in Appendix A.

**IODINE & GENESIS.** In the 3D-Room environment, IODINE is able to segment the objects and the background into separate components. However, it occasionally does not properly decompose objects (in the Large 3D-room results, the orange sphere on the right is not reconstructed) and may generate blurry objects. GENESIS is able to segment the background walls, floor, and sky into multiple components. It is able to capture blurry foreground objects in the Small 3D-Room, but is not able to cleanly capture foreground objects with the larger number of objects in the Large 3D-Room. In Atari, both IODINE and GENESIS fail to capture the foreground properly or try to encode all objects in a single component. We believe this is because the objects in Atari games are smaller, less regular and lack the obvious latent factors like color and shape as in the 3D dataset, and thus detection-based approaches are more appropriate in this case.

**SPAIR & SPAIR-P.** SPAIR is able to detect tight bounding boxes in both 3D-Rooms and the two Atari games. SPAIR-P, however, often fails to detect the foreground objects in proper bounding boxes, frequently uses multiple bounding boxes for one object and redundantly detects parts of the background as foreground objects. This is a limitation of the patch training as the receptive field of

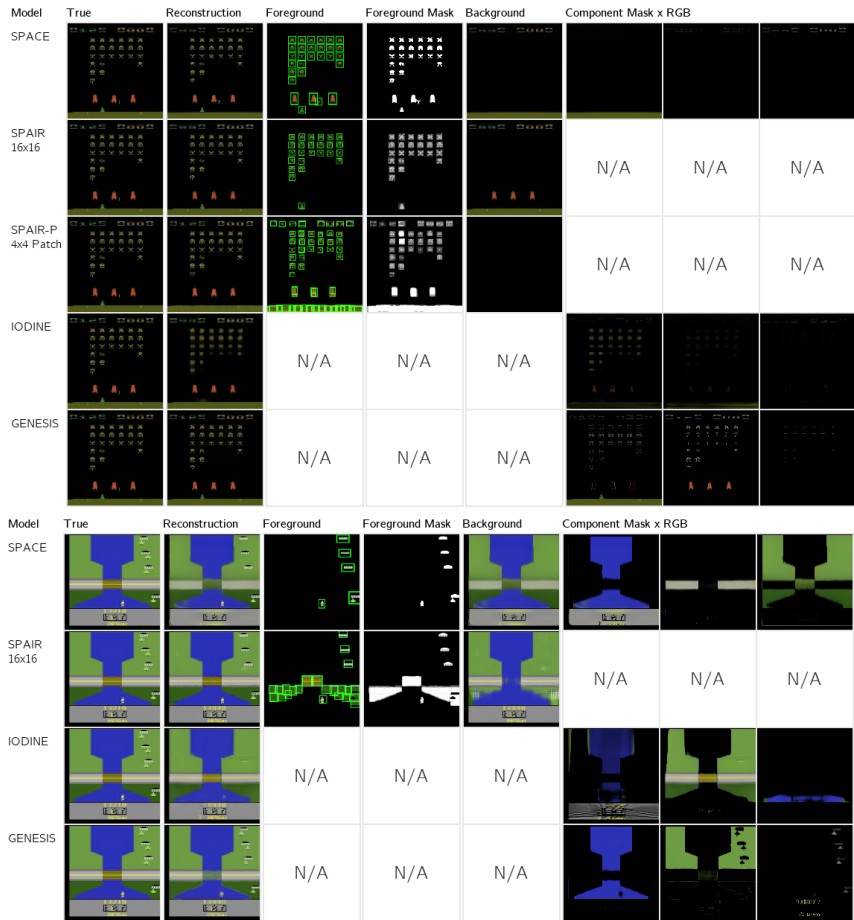

**Figure 3:** Qualitative comparison between SPACE , SPAIR, IODINE and GENESIS for Space Invaders, Air Raid, and River Raid.

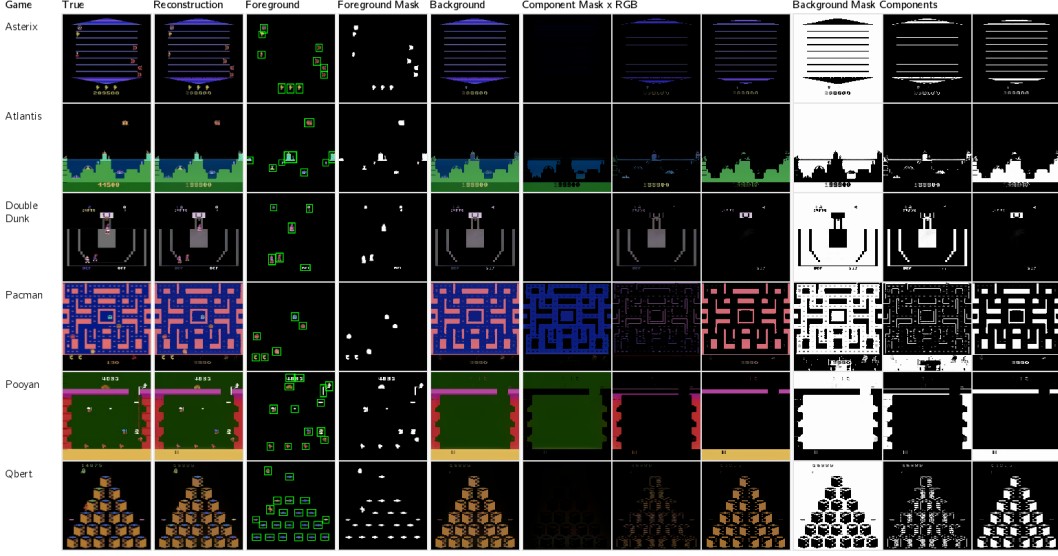

**Figure 4:** Qualitative demonstration of SPACE trained jointly on a selection of 10 Atari games. We show 6 games with complex background here.

each patch is limited to a $32 \times 32$ glimpse, prohibiting it from detecting larger objects and making it difficult to distinguish the background from foreground. These two properties are illustrated well in Space Invaders, where it is able to detect the small aliens, but it detects the long piece of background ground on the bottom of the image as foreground objects.

**SPACE.** In both 3D-Room, SPACE is able to accurately detect almost all objects despite the large variations in object positions, colors, and shapes, while producing a clean segmentation of the background walls, ground, and sky. This is in contrast to the SPAIR model, while being able to provide similar foreground detection quality, encodes the whole background into a single component, which makes the representation less disentangled. Notably, in River Raid where the background is constantly changing, SPACE is able to perfectly segment the blue river while accurately detecting all foreground objects, while SPAIR often cannot properly separate foreground and background.

**Joint Training.** Figure 4 shows the results of training SPACE jointly across 10 Atari games. We see that even in this setting, SPACE is able to correctly detect foreground objects and cleanly segment the background.

**Foreground vs Background.** Typically, foreground is the dynamic local part of the scene that we are interested in, and background is the relatively static and global part. This definition, though intuitive, is ambiguous. Some objects, such as the red shields in Space Invaders and the key in Montezuma's Revenge (Figure 6) are static but important to detect as foreground objects. We found that SPACE tends to detect these as foreground objects while SPAIR considers it background. Similar behavior is observed in Atlantis (Figure 4), where SPACE tends to detect some foreground objects from the middle base that is above the water. One reason for this behavior is because we limit the capacity of the background module by using a spatial broadcast decoder (Watters et al., 2019) which is much weaker when compared to other decoders like sub-pixel convolutional nets (Shi et al. (2016)). This would favor modeling static objects as foreground rather than background.

## 4.2 QUANTITATIVE COMPARISON

In this section we compare SPACE with the baselines in several quantitative metrics[2]. We first note that each of the baseline models has a different *decomposition capacity* ($\mathcal{C}$), which we define as the capability of the model to decompose the scene into its semantic constituents such as the foreground objects and the background segmented components. For SPACE, the decomposition capacity is equal to the number of grid cells $H \times W$ (which is the maximum number of foreground objects that can be detected) plus the number of background components $K$. For SPAIR, the decomposition capacity is equal to the number of grid cells $H \times W$ plus 1 for background. For IODINE and GENESIS, it is equal to the number of components $K$.

For each experiment, we compare the metrics for each model with similar decomposition capacities. This way, each model can decompose the image into the same number of components. For a setting in SPACE with a grid size of $H \times W$ with $K_{\text{SPACE}}$ components, the equivalent settings in IODINE and GENESIS would be with $\mathcal{C} = (H \times W) + K_{\text{SPACE}}$. The equivalent setting in SPAIR would be a grid size of $H \times W$.

**Table 1:** Comparison of SPACE the SPAIR baseline with respect to the quality of the bounding boxes in the 3D-Room setting. Results are averaged over 5 best random seeds and standard deviations are given.

| Model | Dataset | Avg. Precision IoU Threshold $= 0.5$ | Avg. Precision IoU Threshold $\in [0.5 : 0.05 : 0.95]$ | Object Count Error Rate |
|---|---|---|---|---|
| SPACE ($16 \times 16$) | 3D-Room Large | $0.8927 \pm 0.0027$ | $0.4445 \pm 0.0075$ | $0.0446 \pm 0.0026$ |
| SPAIR ($16 \times 16$) | 3D-Room Large | $0.9072 \pm 0.0003$ | $0.4364 \pm 0.0179$ | $0.0360 \pm 0.0072$ |
| SPACE ($8 \times 8$) | 3D-Room Small | $0.9027 \pm 0.0009$ | $0.5069 \pm 0.0030$ | $0.0397 \pm 0.0026$ |
| SPAIR ($8 \times 8$) | 3D-Room Small | $0.9081 \pm 0.0004$ | $0.5068 \pm 0.0081$ | $0.0209 \pm 0.0039$ |

**Gradient Step Latency.** The leftmost chart of Figure 5 shows the time taken to complete one gradient step (forward and backward propagation) for different decomposition capacities for each of the models. We see that SPAIR's latency grows with the number of cells because of the sequential

---

[2]As previously shown, SPAIR-P does not work well in many of our environments, so we do not include it in these quantitative experiments. As in the qualitative section, we use the SPAIR implementation with sequential inference and parallel rendering in order to speed up the experiments.

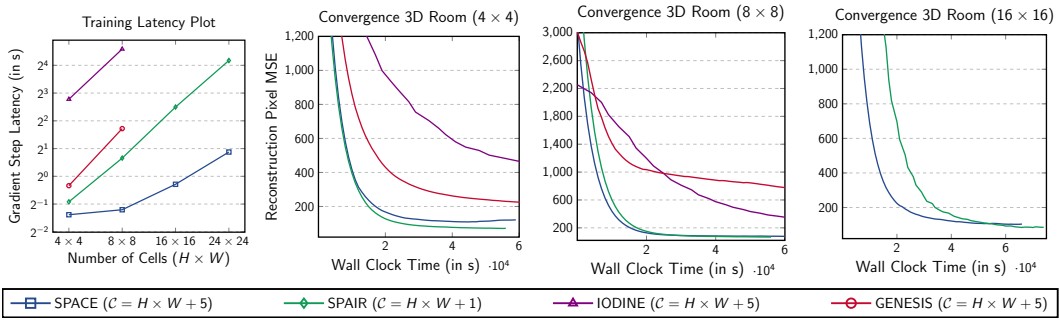

**Figure 5:** Quantitative performance comparison between SPACE , SPAIR, IODINE and GENESIS in terms of batch-processing time during training, training convergence and converged pixel MSE. Convergence plots showing pixel-MSE were computed on a held-out set during training.

nature of its latent inference step. Similarly GENESIS and IODINE's latency grows with the number of components $K$ because each component is processed sequentially in both the models. IODINE is the slowest overall with its computationally expensive iterative inference procedure. Furthermore, both IODINE and GENESIS require storing data for each of the $K$ components, so we were unable to run our experiments on 256 components or greater before running out of memory on our 22GB GPU. On the other hand, SPACE employs parallel processing for the foreground which makes it scalable to large grid sizes, allowing it to detect a large number of foreground objects without any significant performance degradation. Although this data was collected for gradient step latency, this comparison implies a similar relationship exists with inference time which is a main component in the gradient step.

**Time for Convergence.** The remaining three charts in Figure 5 show the amount of time each model takes to converge in different experimental settings. We use the pixel-wise mean squared error (MSE) as a measurement of how close a model is to convergence. In all settings, SPAIR and SPACE converge much faster than IODINE and GENESIS. In the $4 \times 4$ and $8 \times 8$ setting, SPAIR and SPACE converge equally fast. But as we scale up to $16 \times 16$, SPAIR becomes much slower than SPACE .

**Average Precision and Error Rate.** In order to assess the quality of our bounding box predictions, we measure the Average Precision and Object Count Error Rate of our predictions. Our results are shown in Table 1. We only report these metrics for 3D-Room since we have access to the ground truth bounding boxes for each of the objects in the scene. Both models have very similar average precision and error rate. Despite being parallel in its inference, SPACE has a comparable count error rate to that of SPAIR.

From our experiments, we can assert that SPACE can produce similar quality bounding boxes as SPAIR while 1) having orders of magnitude faster inference and gradient step time, 2) scaling to a large number of objects without significant performance degradation, and 3) providing complex background segmentation.

## 5 CONCLUSION

We propose SPACE, a unified probabilistic model that combines the benefits of the object representation models based on spatial attention and the scene decomposition models based on component mixture. SPACE can explicitly provide factorized object representation per foreground object while also decomposing complex background segments. SPACE also achieves a significant speed-up and thus makes the model applicable to scenes with a much larger number of objects without performance degradation. Besides, the detected objects in SPACE are also more intuitive than other methods. We show the above properties of SPACE on Atari and 3D-Rooms. Interesting future directions are to replace the sequential processing of background by a parallel one and to improve the model for natural images. Our next plan is to apply SPACE for object-oriented model-based reinforcement learning.

ACKNOWLEDGMENTS

SA thanks to Kakao Brain and Center for Super Intelligence (CSI) for their support. ZL thanks to the ZJU-3DV group for its support. The authors would also like to thank Chang Chen and Zhuo Zhi for their insightful discussions and help in generating the 3D-Room dataset.

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

# A  ADDITIONAL RESULTS OF SPACE

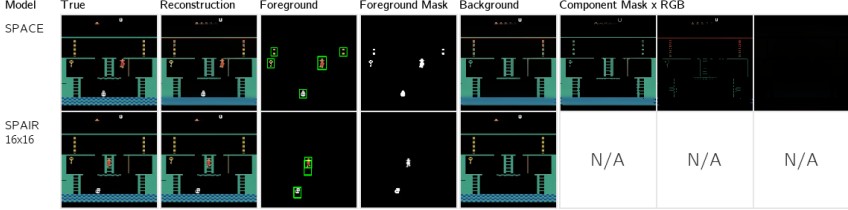

**Figure 6:** Case illustration of Montezuma's Revenge comparing object-detection behaviour in SPACE and SPAIR.

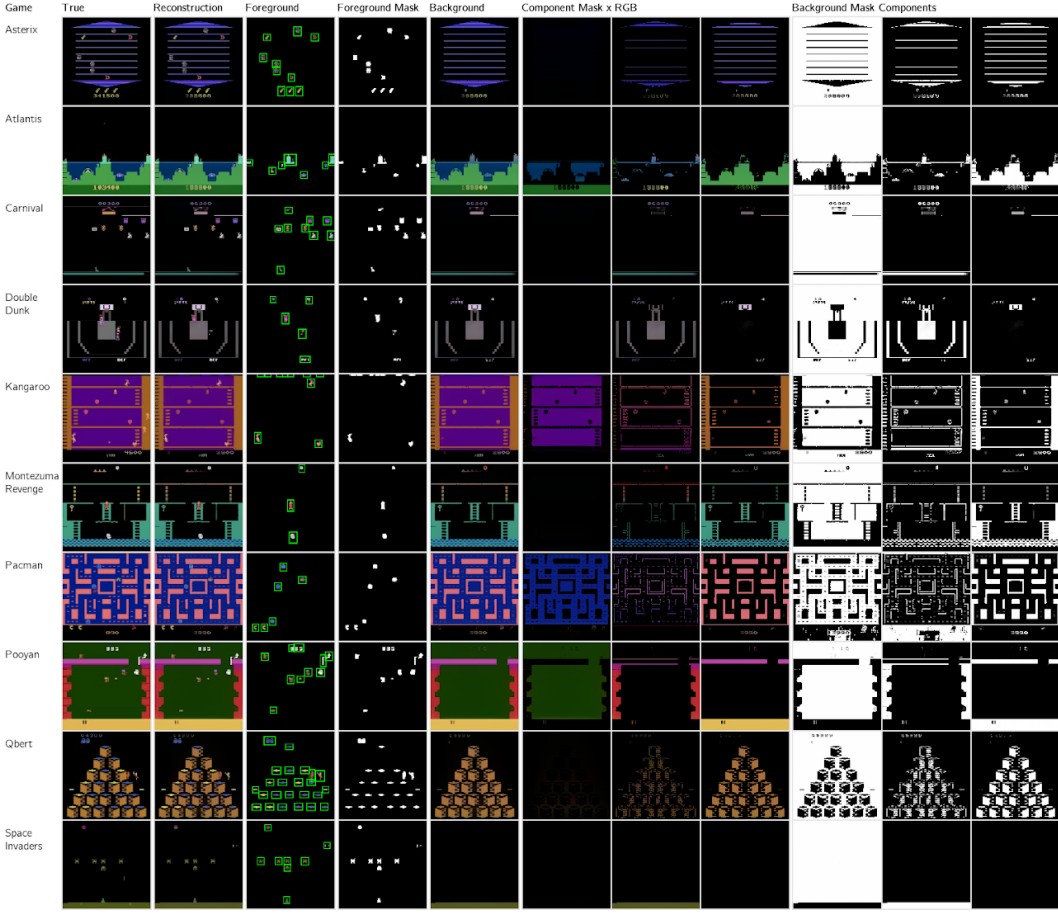

**Figure 7:** Qualitative demonstration of SPACE trained on the jointly on a selection of 10 ATARI games.

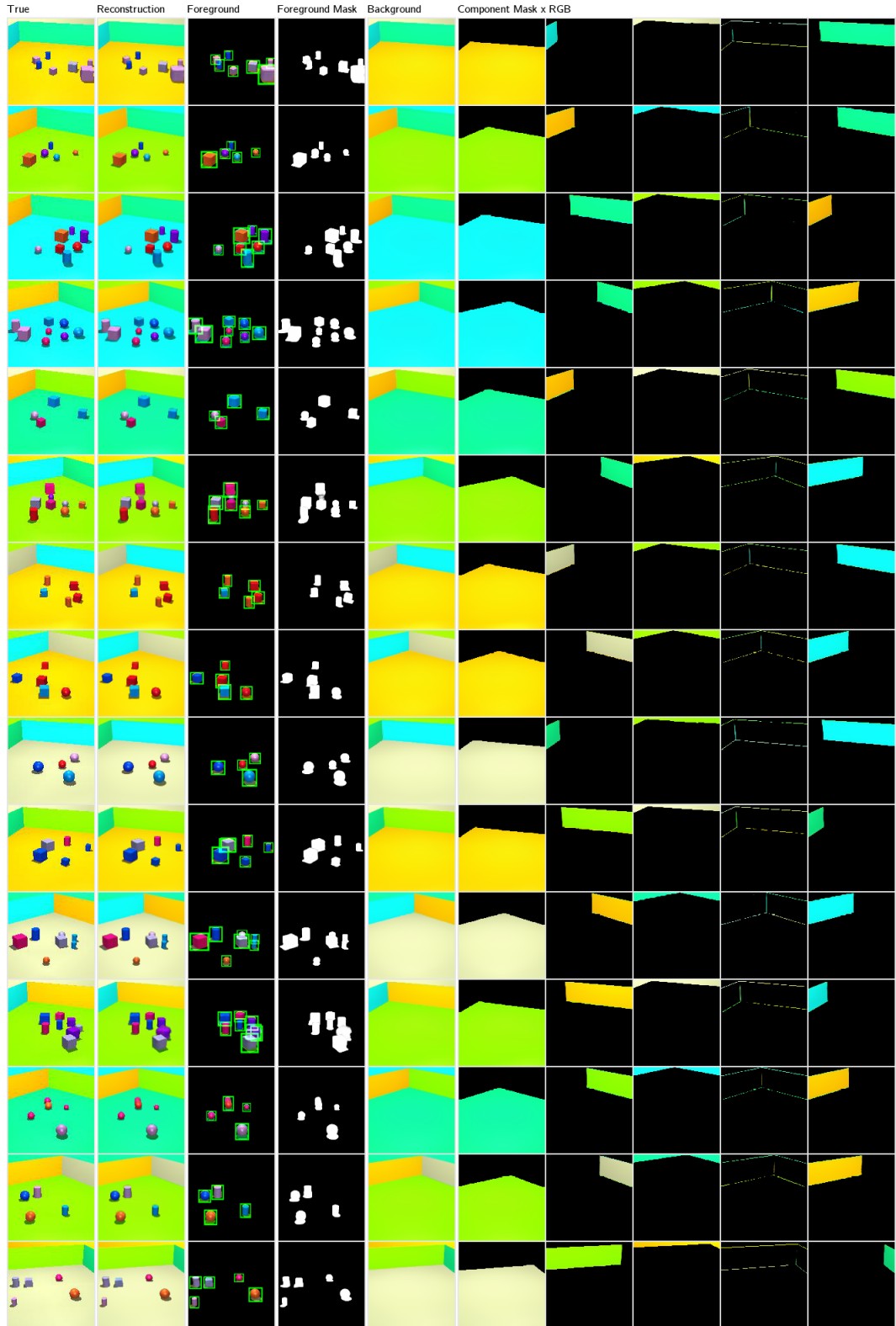

**Figure 8:** Object detection and background segmentation using SPACE on 3D-Room data set with small number of objects. Each row corresponds to one input image.

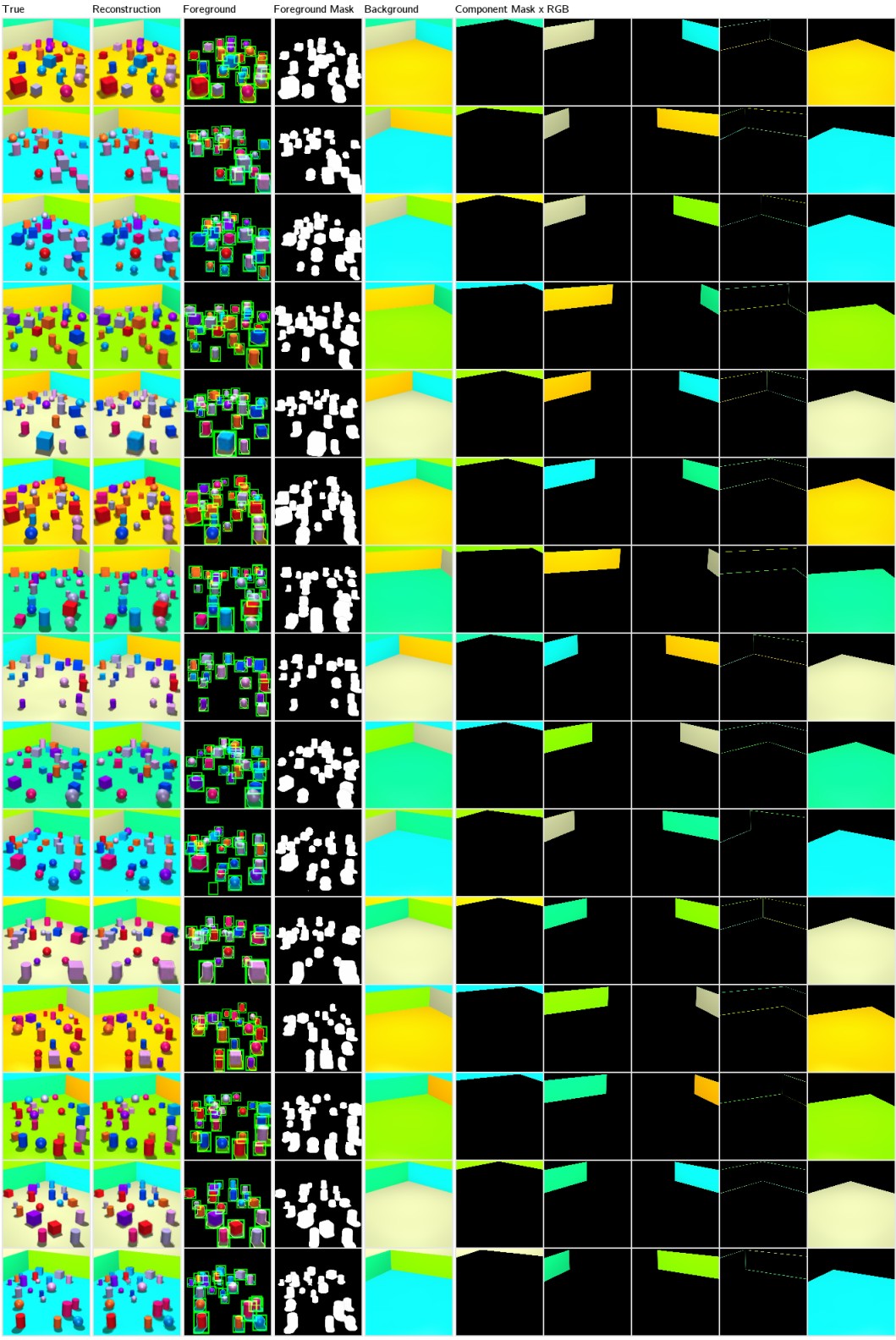

**Figure 9:** Object detection and background segmentation using SPACE on 3D-Room data set with large number of objects.

# B  ELBO DERIVATIONS

In this section, we derive the ELBO for the log-likelihood $\log p(\mathbf{x})$.

$$
\begin{aligned}
\log p(\mathbf{x}) &\geq \mathbb{E}_{q(\mathbf{z}^{\mathrm{fg}}, \mathbf{z}^{\mathrm{bg}} | \mathbf{x})} \left[ p(\mathbf{x} | \mathbf{z}^{\mathrm{fg}}, \mathbf{z}^{\mathrm{bg}}) \right] - D_{\mathrm{KL}}(q(\mathbf{z}^{\mathrm{fg}} | \mathbf{x}) \,\|\, p(\mathbf{z}^{\mathrm{fg}})) - D_{\mathrm{KL}}(q(\mathbf{z}^{\mathrm{bg}} | \mathbf{x}) \,\|\, p(\mathbf{z}^{\mathrm{bg}})) \\
&= \mathbb{E}_{q(\mathbf{z}^{\mathrm{fg}}, \mathbf{z}^{\mathrm{bg}} | \mathbf{x})} \left[ p(\mathbf{x} | \mathbf{z}^{\mathrm{fg}}, \mathbf{z}^{\mathrm{bg}}) - D_{\mathrm{KL}}(q(\mathbf{z}^{\mathrm{fg}} | \mathbf{x}) \,\|\, p(\mathbf{z}^{\mathrm{fg}})) - D_{\mathrm{KL}}(q(\mathbf{z}^{\mathrm{bg}} | \mathbf{x}) \,\|\, p(\mathbf{z}^{\mathrm{bg}})) \right] \\
&= \mathbb{E}_{q(\mathbf{z}^{\mathrm{fg}}, \mathbf{z}^{\mathrm{bg}} | \mathbf{x})} \Big[ p(\mathbf{x} | \mathbf{z}^{\mathrm{fg}}, \mathbf{z}^{\mathrm{bg}}) - \sum_{k=1}^{K} D_{\mathrm{KL}}(q(\mathbf{z}_k^{\mathrm{bg}} | \mathbf{x}, \mathbf{z}_{<k}^{\mathrm{bg}}) \,\|\, p(\mathbf{z}^{\mathrm{bg}} | \mathbf{z}_{<k}^{\mathrm{bg}})) - \\
&\qquad\qquad \sum_{i=1}^{H \times W} D_{\mathrm{KL}}(q(\mathbf{z}_i^{\mathrm{fg}} | \mathbf{x}) \,\|\, p(\mathbf{z}_i^{\mathrm{fg}})) \Big]
\end{aligned}
$$

**KL Divergence for the Foreground Latents** Under the SPACE 's approximate inference, the $D_{\mathrm{KL}}(q(\mathbf{z}_i^{\mathrm{fg}} | \mathbf{x}) \,\|\, p(\mathbf{z}_i^{\mathrm{fg}}))$ inside the expectation can be evaluated as follows.

$$
\begin{aligned}
&\mathbb{E}_{q(\mathbf{z}^{\mathrm{fg}}, \mathbf{z}^{\mathrm{bg}} | \mathbf{x})} \left[ D_{\mathrm{KL}}(q(\mathbf{z}_i^{\mathrm{fg}} | \mathbf{x}) \,\|\, p(\mathbf{z}_i^{\mathrm{fg}})) \right] \\
&= \mathbb{E}_{q(\mathbf{z}^{\mathrm{fg}}, \mathbf{z}^{\mathrm{bg}} | \mathbf{x})} \Bigg[ D_{\mathrm{KL}}(q(\mathbf{z}_i^{\mathrm{pres}} | \mathbf{x}) \,\|\, p(\mathbf{z}_i^{\mathrm{pres}})) + \mathbb{E}_{q(\mathbf{z}_i^{\mathrm{pres}} | \mathbf{x})} \mathbf{z}_i^{\mathrm{pres}} \Big[ D_{\mathrm{KL}}(q(\mathbf{z}_i^{\mathrm{where}} | \mathbf{x}) \,\|\, p(\mathbf{z}_i^{\mathrm{where}})) \\
&\qquad + \mathbb{E}_{q(\mathbf{z}_i^{\mathrm{where}} | \mathbf{x})} D_{\mathrm{KL}}(q(\mathbf{z}_i^{\mathrm{what}} | \mathbf{x}, \mathbf{z}_i^{\mathrm{where}}) \,\|\, p(\mathbf{z}_i^{\mathrm{what}})) + D_{\mathrm{KL}}(q(\mathbf{z}_i^{\mathrm{depth}} | \mathbf{x}) \,\|\, p(\mathbf{z}_i^{\mathrm{depth}})) \Big] \Bigg] \\
&= \mathbb{E}_{q(\mathbf{z}^{\mathrm{fg}}, \mathbf{z}^{\mathrm{bg}} | \mathbf{x})} \Bigg[ D_{\mathrm{KL}}(q(\mathbf{z}_i^{\mathrm{pres}} | \mathbf{x}) \,\|\, p(\mathbf{z}_i^{\mathrm{pres}})) + \mathbf{z}_i^{\mathrm{pres}} \Big[ D_{\mathrm{KL}}(q(\mathbf{z}_i^{\mathrm{where}} | \mathbf{x}) \,\|\, p(\mathbf{z}_i^{\mathrm{where}})) \\
&\qquad + D_{\mathrm{KL}}(q(\mathbf{z}_i^{\mathrm{what}} | \mathbf{x}, \mathbf{z}_i^{\mathrm{where}}) \,\|\, p(\mathbf{z}_i^{\mathrm{what}})) + D_{\mathrm{KL}}(q(\mathbf{z}_i^{\mathrm{depth}} | \mathbf{x}) \,\|\, p(\mathbf{z}_i^{\mathrm{depth}})) \Big] \Bigg]
\end{aligned}
$$

**KL Divergence for the Background Latents** Under our GENESIS-like modeling of inference for the background latents, the KL term inside the expectation for the background is evaluated as follows.

$$
\begin{aligned}
&\mathbb{E}_{q(\mathbf{z}^{\mathrm{fg}}, \mathbf{z}^{\mathrm{bg}} | \mathbf{x})} \left[ D_{\mathrm{KL}}(q(\mathbf{z}_k^{\mathrm{bg}} | \mathbf{z}_{<k}^{\mathrm{bg}}, \mathbf{x}) \,\|\, p(\mathbf{z}_k^{\mathrm{bg}} | \mathbf{z}_{<k}^{\mathrm{bg}})) \right] \\
&= \mathbb{E}_{q(\mathbf{z}^{\mathrm{fg}}, \mathbf{z}^{\mathrm{bg}} | \mathbf{x})} \left[ D_{\mathrm{KL}}(q(\mathbf{z}_k^{m} | \mathbf{z}_{<k}^{m}, \mathbf{x}) \,\|\, p(\mathbf{z}_k^{m} | \mathbf{z}_{<k}^{m})) + \mathbb{E}_{q(\mathbf{z}_k^{m} | \mathbf{z}_{<k}^{m}, \mathbf{x})} D_{\mathrm{KL}}(q(\mathbf{z}_k^{c} | \mathbf{z}_k^{m}, \mathbf{x}) \,\|\, p(\mathbf{z}_k^{c} | \mathbf{z}_k^{m})) \right] \\
&= \mathbb{E}_{q(\mathbf{z}^{\mathrm{fg}}, \mathbf{z}^{\mathrm{bg}} | \mathbf{x})} \left[ D_{\mathrm{KL}}(q(\mathbf{z}_k^{m} | \mathbf{z}_{<k}^{m}, \mathbf{x}) \,\|\, p(\mathbf{z}_k^{m} | \mathbf{z}_{<k}^{m})) + D_{\mathrm{KL}}(q(\mathbf{z}_k^{c} | \mathbf{z}_k^{m}, \mathbf{x}) \,\|\, p(\mathbf{z}_k^{c} | \mathbf{z}_k^{m})) \right]
\end{aligned}
$$

**Relaxed treatment of $\mathbf{z}_i^{\mathrm{pres}}$** In our implementation, we model the Bernoulli random variable $\mathbf{z}_i^{\mathrm{pres}}$ using the Gumbel-Softmax distribution (Jang et al., 2016). We use the relaxed value of $\mathbf{z}^{\mathrm{pres}}$ in the entire training and use hard samples only for the visualizations.

# C  BOUNDARY LOSS

In this section we elaborate on the implementation details of the *boundary loss*. We construct a kernel of the size of the glimpse, $gs \times gs$ (we use $gs = 32$) with a boundary gap of $b = 6$ having negative uniform weights inside the boundary and a zero weight in the region between the boundary and the glimpse. This ensures that the model is penalized when the object is outside the boundary. This kernel is first mapped onto the global space via STN Jaderberg et al. (2015) to obtain the global kernel. This is then multiplied element-wise with global object mask $\alpha$ to obtain the *boundary loss map*. The objective of the loss is to minimize the mean of this *boundary loss map*. In addition to the ELBO, this loss is also back-propagated via *RMSProp* (Tieleman & Hinton. (2012)). This loss, due to the boundary constraint, enforces the bounding boxes to be less tight and results in lower average precision, so we disable the loss and optimize only the ELBO after the model has converged well.

# D IMPLEMENTATION DETAILS

## D.1 ALGORITHMS

Algorithm 1 and Algorithm 3 present SPACE's inference for foreground and background. Algorithm 2 describe the details of $\text{rescale}_i$ function in Algorithm 1 that transforms local shift $\mathbf{z}_i^{\text{shift}}$ to global shift $\hat{\mathbf{z}}_i^{\text{shift}}$.Algorithm 4 show the details of the generation process of the background module. For foreground generation, we simply sample the latent variables from the priors instead of conditioning on the input. Note that, for convenience the algorithms for the foreground module and background module are presented with `for` loops, but inference for all variables of the foreground module are implemented as parallel convolution operations and most operations of the background module (barring the LSTM module) are parallel as well.

---

**Algorithm 1:** Foreground Inference

---

**Input:** image $\boldsymbol{x}$
**Output:** foreground mask $\alpha$, appearance $\mu^{\text{fg}}$, grid height $H$ and width $W$
$\hat{e}^{\text{img}} = \text{ImageEncoderFg}(\boldsymbol{x})$
$\boldsymbol{r}^{\text{img}} = \text{ResidualConnection}(\hat{e}^{\text{img}})$
$\boldsymbol{e}^{\text{img}} = \text{ResidualEncoder}([\hat{e}^{\text{img}}, \boldsymbol{r}^{\text{img}}])$
**for** $i \leftarrow 1$ **to** $HW$ **do**
  /* The following is performed in parallel         */
  $\boldsymbol{\rho}_i = \text{ZPresNet}(\boldsymbol{e}_i^{\text{img}})$
  $[\boldsymbol{\mu}_i^{\text{depth}}, \boldsymbol{\sigma}_i^{\text{depth}}] = \text{ZDepthNet}(\boldsymbol{e}_i^{\text{img}})$
  $[\boldsymbol{\mu}_i^{\text{scale}}, \boldsymbol{\sigma}_i^{\text{scale}}] = \text{ZScaleNet}(\boldsymbol{e}_i^{\text{img}})$
  $[\boldsymbol{\mu}_i^{\text{shift}}, \boldsymbol{\sigma}_i^{\text{shift}}] = \text{ZShiftNet}(\boldsymbol{e}_i^{\text{img}})$
  $\mathbf{z}_i^{\text{pres}} \sim \text{Bern}(\boldsymbol{\rho}_i)$
  $\mathbf{z}_i^{\text{depth}} \sim \mathcal{N}(\boldsymbol{\mu}_i^{\text{depth}}, \boldsymbol{\sigma}_i^{\text{depth},2})$
  $\mathbf{z}_i^{\text{scale}} \sim \mathcal{N}(\boldsymbol{\mu}_i^{\text{scale}}, \boldsymbol{\sigma}_i^{\text{scale},2})$
  $\mathbf{z}_i^{\text{shift}} \sim \mathcal{N}(\boldsymbol{\mu}_i^{\text{shift}}, \boldsymbol{\sigma}_i^{\text{shift},2})$
  /* Rescale local shift to global shift as in SPAIR      */
  $\hat{\mathbf{z}}_i^{\text{scale}} = \sigma(\mathbf{z}_i^{\text{scale}})$
  $\hat{\mathbf{z}}_i^{\text{shift}} = \text{rescale}_i(\mathbf{z}_i^{\text{shift}})$
  $\mathbf{z}_i^{\text{where}} = [\hat{\mathbf{z}}_i^{\text{scale}}, \hat{\mathbf{z}}_i^{\text{shift}}]$
  /* Extract glimpses with a Spatial Transformer       */
  $\hat{\boldsymbol{x}}_i = \text{ST}(\boldsymbol{x}, \mathbf{z}_i^{\text{where}})$
  $[\boldsymbol{\mu}_i^{\text{what}}, \boldsymbol{\sigma}_i^{\text{what}}] = \text{GlimpseEncoder}(\hat{\boldsymbol{x}}_i)$
  $\mathbf{z}_i^{\text{what}} \sim \mathcal{N}(\boldsymbol{\mu}_i^{\text{what}}, \boldsymbol{\sigma}_i^{\text{what},2})$
  /* Foreground mask and appearance of glimpse size     */
  $[\boldsymbol{\alpha}_i^{\text{att}}, \boldsymbol{o}_i^{\text{att}}] = \text{GlimpseDecoder}(\mathbf{z}_i^{\text{what}})$
  $\hat{\boldsymbol{\alpha}}_i^{\text{att}} = \boldsymbol{\alpha}_i^{\text{att}} \odot \mathbf{z}_i^{\text{pres}}$
  $\boldsymbol{y}_i^{\text{att}} = \hat{\boldsymbol{\alpha}}_i^{\text{att}} \odot \boldsymbol{o}_i^{\text{att}}$
  /* Transform both to canvas size            */
  $\hat{\boldsymbol{\alpha}}_i^{\text{att}} = \text{ST}^{-1}(\hat{\boldsymbol{\alpha}}_i^{\text{att}}, \mathbf{z}_i^{\text{where}})$
  $\boldsymbol{y}_i^{\text{att}} = \text{ST}^{-1}(\boldsymbol{y}_i^{\text{att}}, \mathbf{z}_i^{\text{where}})$
**end**
/* Compute weights for each component           */
$\boldsymbol{w} = \text{softmax}(-100 \cdot \sigma(\mathbf{z}^{\text{depth}}) \odot \hat{\boldsymbol{\alpha}}^{\text{att}})$
/* Compute global weighted mask and foreground appearance   */
$\alpha = \text{sum}(\boldsymbol{w} \odot \hat{\boldsymbol{\alpha}}^{\text{att}})$
$\mu^{\text{fg}} = \text{sum}(\boldsymbol{w} \odot \boldsymbol{y}^{\text{att}})$

---

---

**Algorithm 2:** Rescale $\mathbf{z}_i^{\text{shift}}$

---

**Input:** Shift latent $\mathbf{z}_i^{\text{shift}}$, cell index $i$, grid height $H$ and width $W$
**Output:** Rescaled shift latent $\hat{\mathbf{z}}_i^{\text{shift}}$
```
/* Get width and heigh index of cell i                          */
```
$[k, j] = [i\%H, i \div H]$
```
/* Center of this cell                                          */
```
$\mathbf{c}_i = [k + 0.5, j + 0.5]$
```
/* Get global shift                                             */
```
$\tilde{\mathbf{z}}_i^{\text{shift}} = \mathbf{c}_i + \tanh(\mathbf{z}_i^{\text{shift}})$
```
/* Normalize to range (-1, 1)                                   */
```
$\hat{\mathbf{z}}_i^{\text{shift}} = 2 \cdot \tilde{\mathbf{z}}_i^{\text{shift}}/[W, H] - 1$

---

**Algorithm 3:** Background Inference

---

**Input:** image $\boldsymbol{x}$, initial LSTM states $\boldsymbol{h}_0, \boldsymbol{c}_0$, initial dummy mask $\mathbf{z}_0^m$
**Output:** background masks $\pi_k$, appearance $\mu_k^{\text{bg}}$, for $k = 1, \dots, K$
$\boldsymbol{e}^{\text{img}} = \text{ImageEncoderBg}(\boldsymbol{x})$
**for** $k \leftarrow 1$ **to** $K$ **do**
    $\boldsymbol{h}_k, \boldsymbol{c}_k = \text{LSTM}([\mathbf{z}_{k-1}^m, \boldsymbol{e}^{\text{img}}], \boldsymbol{c}_{k-1}, \boldsymbol{h}_{k-1})$
    $[\boldsymbol{\mu}_k^m, \boldsymbol{\sigma}_k^m] = \text{PredictMask}(\boldsymbol{h}_k)$
    $\mathbf{z}^m \sim \mathcal{N}(\boldsymbol{\mu}_k^m, \boldsymbol{\sigma}_k^{m,2})$
    `/* Actually decoded in parallel                        */`
    $\hat{\pi}_k = \text{MaskDecoder}(\mathbf{z}_k^m)$
    `/* Stick breaking process as described in GENESIS       */`
    $\pi_k = \text{SBP}(\hat{\pi}_{1:k})$
    $[\boldsymbol{\mu}_k^c, \boldsymbol{\sigma}_k^c] = \text{CompEncoder}([\pi_k, \boldsymbol{x}])$
    $\mathbf{z}^c \sim \mathcal{N}(\boldsymbol{\mu}_k^c, \boldsymbol{\sigma}_k^{c,2})$
    $\mu_k^{\text{bg}} = \text{CompDecoder}(\mathbf{z}_k^c)$
**end**

---

**Algorithm 4:** Background Generation

---

**Input:** initial LSTM states $\boldsymbol{h}_0, \boldsymbol{c}_0$, initial dummy mask $\mathbf{z}_0^m$
**Output:** background masks $\pi_k$, appearance $\mu_k^{\text{bg}}$, for $k = 1, \dots, K$
**for** $k \leftarrow 1$ **to** $K$ **do**
    $\boldsymbol{h}_k, \boldsymbol{c}_k = \text{LSTM}(\mathbf{z}_{k-1}^m, \boldsymbol{c}_{k-1}, \boldsymbol{h}_{k-1})$
    $[\boldsymbol{\mu}_k^m, \boldsymbol{\sigma}_k^m] = \text{PredictMaskPrior}(\boldsymbol{h}_k)$
    $\mathbf{z}^m \sim \mathcal{N}(\boldsymbol{\mu}_k^m, \boldsymbol{\sigma}_k^{m,2})$
    `/* Actually decoded in parallel                        */`
    $\hat{\pi}_k = \text{MaskDecoder}(\mathbf{z}_k^m)$
    `/* Stick breaking process as described in GENESIS       */`
    $\pi_k = \text{SBP}(\hat{\pi}_{1:k})$
    $[\boldsymbol{\mu}_k^c, \boldsymbol{\sigma}_k^c] = \text{PredictComp}(\mathbf{z}_k^m)$
    $\mathbf{z}_k^c \sim \mathcal{N}(\boldsymbol{\mu}_k^c, \boldsymbol{\sigma}_k^{c,2})$
    $\mu_k^{\text{bg}} = \text{CompDecoder}(\mathbf{z}_k^c)$
**end**

---

## D.2 Training Regime and Hyperparameters

For all experiments we use an image size of $128 \times 128$ and a batch size of 12 to 16 depending on memory usage. For the foreground module, we use the RMSProp (Tieleman & Hinton. (2012)) optimizer with a learning rate of $1 \times 10^{-5}$ except for Figure 5, for which we use a learning rate of $1 \times 10^{-4}$ as SPAIR. For the background module, we use the Adam (Kingma & Ba (2014)) optimizer with a learning rate of $1 \times 10^{-3}$. We use gradient clipping with a maximum norm of 1.0. For quantitative results, SPACE is trained up to 160000 steps. For Atari games, we find it beneficial to set $\alpha$ to be fixed for the first several thousand steps, and vary the actual value and number of steps for different games. This allows both the foreground as well as the background module to learn in the early stage of training.

We list out our hyperparameters for 3D large dataset and joint training for 10 static Atari games below. Hyperparameters for other experiments are similar, but are finetuned for each dataset individually. In the tables below, $(m \rightarrow n) : (p \rightarrow q)$ denotes annealing the hyperparameter value from $m$ to $n$, starting from step $p$ until step $q$.

**3D Room Large**

| Name | Symbol | Value |
|------|--------|-------|
| $\mathbf{z}^{\text{pres}}$ prior prob | $\boldsymbol{\rho}$ | $(0.1 \rightarrow 0.01) : (4000 \rightarrow 10000)$ |
| $\mathbf{z}^{\text{scale}}$ prior mean | $\boldsymbol{\mu}^{\text{scale}}$ | $(-1.0 \rightarrow -2.0) : (10000 \rightarrow 20000)$ |
| $\mathbf{z}^{\text{scale}}$ prior stdev | $\boldsymbol{\sigma}^{\text{scale}}$ | 0.1 |
| $\mathbf{z}^{\text{shift}}$ prior | $\boldsymbol{\mu}^{\text{shift}}, \boldsymbol{\sigma}^{\text{shift}}$ | $\mathcal{N}(\mathbf{0}, \boldsymbol{I})$ |
| $\mathbf{z}^{\text{depth}}$ prior | $\boldsymbol{\mu}^{\text{depth}}, \boldsymbol{\sigma}^{\text{depth}}$ | $\mathcal{N}(\mathbf{0}, \boldsymbol{I})$ |
| $\mathbf{z}^{\text{what}}$ prior | $\boldsymbol{\mu}^{\text{what}}, \boldsymbol{\sigma}^{\text{what}}$ | $\mathcal{N}(\mathbf{0}, \boldsymbol{I})$ |
| foreground stdev | $\sigma^{\text{fg}}$ | 0.15 |
| background stdev | $\sigma^{\text{bg}}$ | 0.15 |
| component number | $K$ | 5 |
| gumbel-softmax temperature | $\tau$ | $(2.5 \rightarrow 0.5) : (0 \rightarrow 20000)$ |
| #steps to fix $\alpha$ | | N/A |
| fixed $\alpha$ value | | N/A |
| boundary loss | | Yes |
| turn off boundary loss at step | | 100000 |

**Joint Training on 10 Atari Games**

| Name | Symbol | Value |
|------|--------|-------|
| $\mathbf{z}^{\text{pres}}$ prior prob | $\boldsymbol{\rho}$ | $1 \times 10^{-10}$ |
| $\mathbf{z}^{\text{scale}}$ prior mean | $\boldsymbol{\mu}^{\text{scale}}$ | $-2.5$ |
| $\mathbf{z}^{\text{scale}}$ prior stdev | $\boldsymbol{\sigma}^{\text{scale}}$ | 0.1 |
| $\mathbf{z}^{\text{shift}}$ prior | $\boldsymbol{\mu}^{\text{shift}}, \boldsymbol{\sigma}^{\text{shift}}$ | $\mathcal{N}(\mathbf{0}, \boldsymbol{I})$ |
| $\mathbf{z}^{\text{depth}}$ prior | $\boldsymbol{\mu}^{\text{depth}}, \boldsymbol{\sigma}^{\text{depth}}$ | $\mathcal{N}(\mathbf{0}, \boldsymbol{I})$ |
| $\mathbf{z}^{\text{what}}$ prior | $\boldsymbol{\mu}^{\text{what}}, \boldsymbol{\sigma}^{\text{what}}$ | $\mathcal{N}(\mathbf{0}, \boldsymbol{I})$ |
| foreground stdev | $\sigma^{\text{fg}}$ | 0.20 |
| background stdev | $\sigma^{\text{bg}}$ | 0.10 |
| component number | $K$ | 3 |
| gumbel-softmax temperature | $\tau$ | $(2.5 \rightarrow 1.0) : (0 \rightarrow 10000)$ |
| #steps to fix $\alpha$ | | 4000 |
| fixed $\alpha$ value | | 0.1 |
| boundary loss | | No |
| turn off boundary loss at step | | N/A |

## D.3 Model Architecture

Here we describe the architecture of our $16 \times 16$ SPACE model. The model for $8 \times 8$ grid cells is the same but with a stride-2 convolution for the last layer of the image encoder.

All modules that output distribution parameters are implemented with either one single fully connected layer or convolution layer, with the appropriate output size. Image encoders are fully convolutional networks that output a feature map of shape $H \times W$, and the glimpse encoder comprises of convolutional layers followed by a final linear layer that computes the parameters of a Gaussian distribution. For the glimpse decoder of the foreground module and the mask decoder of the background module we use the sub-pixel convolution layer (Shi et al. (2016)). On the lines of GENESIS (Engelcke et al. (2019)) and IODINE (Greff et al. (2019)), we adopt Spatial Broadcast Network (Watters et al. (2019)) as the component decoder to decode $\mathbf{z}_k^c$ into background components.

For inference and generation of the background module, the dependence of $\mathbf{z}_k^m$ on $\mathbf{z}_{1:k-1}^m$ is implemented with LSTMs, with hidden sizes of 64. Dependence of $\mathbf{z}_k^c$ on $\mathbf{z}_k^m$ is implemented with a MLP with two hidden layers with 64 units per layer. We apply softplus when computing standard deviations for all Gaussian distributions, and apply sigmoid when computing reconstruction and masks. We use either Group Normalization (GN) (Wu & He (2018)) and CELU (Barron (2017)) or Batch Normalization (BN) (Ioffe & Szegedy (2015)) and ELU (Clevert et al. (2016)) depending on the module type.

The rest of the architecture details are described below. In the following tables, $\mathrm{ConvSub}(\mathrm{n})$ denotes a sub-pixel convolution layer implemented as a stride-1 convolution and a PyTorch $\mathrm{PixelShuffle}(n)$ layer, and $\mathrm{GN}(n)$ denotes Group Normalization with $n$ groups.

| Name | Value | Comment |
|---|---|---|
| $\mathbf{z}^{\mathrm{depth}}$ dim | 1 | |
| $\mathbf{z}^{\mathrm{pres}}$ dim | 1 | |
| $\mathbf{z}^{\mathrm{scale}}$ dim | 2 | for $x$ and $y$ axis |
| $\mathbf{z}^{\mathrm{shift}}$ dim | 2 | for $x$ and $y$ axis |
| $\mathbf{z}^{\mathrm{what}}$ dim | 32 | |
| $\mathbf{z}^m$ dim | 32 | |
| $\mathbf{z}^c$ dim | 32 | |
| glimpse shape | $(32, 32)$ | for $\mathbf{o}^{\mathrm{att}}, \boldsymbol{\alpha}^{\mathrm{att}}$ |
| image shape | $(128, 128)$ | |

**Foreground Image Encoder**

| Layer | Size/Ch. | Stride | Norm./Act. |
|---|---|---|---|
| Input | 3 | | |
| Conv $4 \times 4$ | 16 | 2 | GN(4)/CELU |
| Conv $4 \times 4$ | 32 | 2 | GN(8)/CELU |
| Conv $4 \times 4$ | 64 | 2 | GN(8)/CELU |
| Conv $3 \times 3$ | 128 | 1 | GN(16)/CELU |
| Conv $3 \times 3$ | 256 | 1 | GN(32)/CELU |
| Conv $1 \times 1$ | 128 | 1 | GN(16)/CELU |

**Residual Connection**

| Layer | Size/Ch. | Stride | Norm./Act. |
|---|---|---|---|
| Input | 128 | | |
| Conv $3 \times 3$ | 128 | 1 | GN(16)/CELU |
| Conv $3 \times 3$ | 128 | 1 | GN(16)/CELU |

**Residual Encoder**

| Layer | Size/Ch. | Stride | Norm./Act. |
|---|---|---|---|
| Input | 128 + 128 | | |
| Conv $3 \times 3$ | 128 | 1 | GN(16)/CELU |

**Glimpse Encoder**

| Layer | Size/Ch. | Stride | Norm./Act. |
|---|---|---|---|
| Input | 3 | | |
| Conv $3 \times 3$ | 16 | 1 | GN(4)/CELU |
| Conv $4 \times 4$ | 32 | 2 | GN(8)/CELU |
| Conv $3 \times 3$ | 32 | 1 | GN(4)/CELU |
| Conv $4 \times 4$ | 64 | 2 | GN(8)/CELU |
| Conv $4 \times 4$ | 128 | 2 | GN(8)/CELU |
| Conv $4 \times 4$ | 256 | 1 | GN(16)/CELU |
| Linear | 32 + 32 | | |

**Glimpse Decoder**

| Layer | Size/Ch. | Stride | Norm./Act. |
|---|---|---|---|
| Input | 32 | | |
| Conv $1 \times 1$ | 256 | 1 | GN(16)/CELU |
| ConvSub(2) | 128 | 1 | GN(16)/CELU |
| Conv $3 \times 3$ | 128 | 1 | GN(16)/CELU |
| ConvSub(2) | 128 | 1 | GN(16)/CELU |
| Conv $3 \times 3$ | 128 | 1 | GN(16)/CELU |
| ConvSub(2) | 64 | 1 | GN(8)/CELU |
| Conv $3 \times 3$ | 64 | 1 | GN(8)/CELU |
| ConvSub(2) | 32 | 1 | GN(8)/CELU |
| Conv $3 \times 3$ | 32 | 1 | GN(8)/CELU |
| ConvSub(2) | 16 | 1 | GN(4)/CELU |
| Conv $3 \times 3$ | 16 | 1 | GN(4)/CELU |

**Background Image Encoder**

| Layer | Size/Ch. | Stride | Norm./Act. |
|---|---|---|---|
| Input | 3 | | |
| Conv $3 \times 3$ | 64 | 2 | BN/ELU |
| Conv $3 \times 3$ | 64 | 2 | BN/ELU |
| Conv $3 \times 3$ | 64 | 2 | BN/ELU |
| Conv $3 \times 3$ | 64 | 2 | BN/ELU |
| Flatten | | | |
| Linear | 64 | | ELU |

**Mask Decoder**

| Layer | Size/Ch. | Stride | Norm./Act. |
|---|---|---|---|
| Input | 32 | | |
| Conv $1 \times 1$ | 256 | 1 | GN(16)/CELU |
| ConvSub(4) | 256 | 1 | GN(16)/CELU |
| Conv $3 \times 3$ | 256 | 1 | GN(16)/CELU |
| ConvSub(2) | 128 | 1 | GN(16)/CELU |
| Conv $3 \times 3$ | 128 | 1 | GN(16)/CELU |
| ConvSub(4) | 64 | 1 | GN(8)/CELU |
| Conv $3 \times 3$ | 64 | 1 | GN(8)/CELU |
| ConvSub(4) | 16 | 1 | GN(4)/CELU |
| Conv $3 \times 3$ | 16 | 1 | GN(4)/CELU |
| Conv $3 \times 3$ | 16 | 1 | GN(4)/CELU |
| Conv $3 \times 3$ | 1 | 1 | |

**Component Encoder**

| Layer | Size/Ch. | Stride | Norm./Act. |
| --- | --- | --- | --- |
| Input | 3+1 (RGB+mask) | | |
| Conv $3 \times 3$ | 32 | 2 | BN/ELU |
| Conv $3 \times 3$ | 32 | 2 | BN/ELU |
| Conv $3 \times 3$ | 64 | 2 | BN/ELU |
| Conv $3 \times 3$ | 64 | 2 | BN/ELU |
| Flatten | | | |
| Linear | 32+32 | | |

**Component Decoder**

| Layer | Size/Ch. | Stride | Norm./Act. |
| --- | --- | --- | --- |
| Input | 32 (1d) | | |
| Spatial Broadcast | 32+2 (3d) | | |
| Conv $3 \times 3$ | 32 | 1 | BN/ELU |
| Conv $3 \times 3$ | 32 | 1 | BN/ELU |
| Conv $3 \times 3$ | 32 | 1 | BN/ELU |
| Conv $3 \times 3$ | 3 | 1 | |

## D.4 BASELINES

Here we give out the details of the background decoder in training of SPAIR (both full image as well as patch-wise training). The foreground and background image encoder is same as that of SPACE with the only difference that the inferred latents are conditioned on previous cells' latents as described in Section 2.2. For the background image encoder, we add an additional linear layer so that the encoded background latent is one dimensional.

**SPAIR Background Decoder**

| Layer | Size/Ch. | Stride | Norm./Act. |
| --- | --- | --- | --- |
| Input | 32 | | |
| Conv $1 \times 1$ | 256 | 1 | GN(16)/CELU |
| ConvSub(4) | 256 | 1 | GN(16)/CELU |
| Conv $3 \times 3$ | 256 | 1 | GN(16)/CELU |
| ConvSub(4) | 128 | 1 | GN(16)/CELU |
| Conv $3 \times 3$ | 128 | 1 | GN(16)/CELU |
| ConvSub(2) | 64 | 1 | GN(8)/CELU |
| Conv $3 \times 3$ | 64 | 1 | GN(8)/CELU |
| ConvSub(4) | 16 | 1 | GN(4)/CELU |
| Conv $3 \times 3$ | 16 | 1 | GN(4)/CELU |
| Conv $3 \times 3$ | 16 | 1 | GN(4)/CELU |
| Conv $3 \times 3$ | 3 | 1 | |

**SPAIR Background Encoder For Patch Training**

| Layer | Size/Ch. | Stride | Norm./Act. |
| --- | --- | --- | --- |
| Input | 3 | | |
| Conv $2 \times 2$ | 16 | 2 | GN(4)/CELU |
| Conv $2 \times 2$ | 32 | 2 | GN(8)/CELU |
| Conv $2 \times 2$ | 64 | 2 | GN(8)/CELU |
| Conv $2 \times 2$ | 128 | 2 | GN(16)/CELU |
| Conv $2 \times 2$ | 32 | 2 | GN(4)/CELU |

**SPAIR Background Decoder For Patch Training**

| Layer | Size/Ch. | Stride | Norm./Act. |
|---|---|---|---|
| Input | 16 | | |
| Conv $1 \times 1$ | 256 | 1 | GN(16)/CELU |
| Conv $1 \times 1$ | 2048 | 1 | |
| ConvSub(4) | 128 | 1 | GN(16)/CELU |
| Conv $3 \times 3$ | 128 | 1 | GN(16)/CELU |
| Conv $1 \times 1$ | 256 | 1 | |
| ConvSub(2) | 64 | 1 | GN(8)/CELU |
| Conv $3 \times 3$ | 64 | 1 | GN(8)/CELU |
| Conv $1 \times 1$ | 256 | 1 | |
| ConvSub(4) | 16 | 1 | GN(4)/CELU |
| Conv $3 \times 3$ | 16 | 1 | GN(4)/CELU |
| Conv $3 \times 3$ | 16 | 1 | GN(4)/CELU |
| Conv $3 \times 3$ | 3 | 1 | |

For IODINE, we use our own implementation following the details as described in (Greff et al., 2019). For GENESIS, we also use our own implementation following the same architecture as in (Engelcke et al., 2019), but the details of individual networks are similar to SPACE's background module.

## E   DATASET DETAILS

**Atari.** For each game, we sample 60,000 random images from a pretrained agent (Wu et al., 2016). We split the images into 50,000 for the training set, 5,000 for the validation set, and 5,000 for the testing set. Each image is preprocessed into a size of $128 \times 128$ pixels with BGR color channels. We present the results for the following games: Space Invaders, Air Raid, River Raid, Montezuma's Revenge.

We also train our model on a dataset of 10 games jointly, where we have 8,000 training images, 1,000 validation images, and 1,000 testing images for each game. We use the following games: Asterix, Atlantis, Carnival, Double Dunk, Kangaroo, Montezuma Revenge, Pacman, Pooyan, Qbert, Space Invaders.

**Room 3D.** We use MuJoCo (Todorov et al., 2012) to generate this dataset. Each image consists of a walled enclosure with a random number of objects on the floor. The possible objects are randomly sized spheres, cubes, and cylinders. The small 3D-Room dataset has 4-8 objects and the large 3D-Room dataset has 18-24 objects. The color of the objects are randomly chosen from 8 different colors and the colors of the background (wall, ground, sky) are chosen randomly from 5 different colors. The angle of the camera is also selected randomly. We use a training set of 63,000 images, a validation set of 7,000 images, and a test set of 7,000 images. We use a 2-D projection from the camera to determine the ground truth bounding boxes of the objects so that we can report the average precision of the different models.

