# OpenReview forum: "SPACE: Unsupervised Object-Oriented Scene Representation via Spatial Attention and Decomposition"
_ICLR.cc/2020/Conference — Accept (Poster)_

### Official Review · AnonReviewer2 · 2019-10-24
**Official Blind Review #2**

**Rating:** 6

**Review:**

Positives:
+The system makes sense and is explained well
+The factoring of scenes into objects and multiple background components is good
+I think overall the experiments are reasonable, although I have a number of questions about whether aspects of them are apples-to-apples

Negatives:
-Some of the experiments do not appear apples-to-apples
-There are a large number of changes, and there aren't any ablations. It's a little hard to follow and verify that the gains are credited properly.

Overall, I'm favorably inclined towards accepting this paper so long as the experiments are more clearly made apples to apples. Right now, since I'm forced to give a binary decision and I'm not positive about comparisons, I have to lean towards rejection -- I'd peg my actual rating as 4.5.

Method:
+The method is well-explained and straight-forward (in a good way).
+The factoring of scenes into objects and multiple background components is good
+The parallelization is good, and the fact that it works far faster than SPAIR with similar results is quite nice

Experiments:
+Overall the experiments are pretty good and compare against the baselines I would expect, and have both qualitative and quantitative results.
+The method appears to do a good job of segmenting the objects, and if Figure 1 is representative, this is quite impressive.
-Why does Figure 1 show results from different systems on different images? This makes comparison impossible. Paired samples are always more informative.

-It's not clear to me that fair comparisons were done, especially to GENESIS.
(a) It's never listed how K for genesis was picked -- this should presumably be tuned somewhere to optimize performance. The paper mentions in the 4.2 that it was impossible to run the experiments for GENESIS for more than 256 components -- but the GENESIS paper has numbers more like K=9. If there are an overabundance of components, this might explain some of the object splitting observed in the paper.
(b) Unless I'm missing something, in Figure 5, for 4x4 and 8x8, it doesn't appear that IODOINE or GENESIS have converged at all. Does the validation MSE just then flatten (or go up) there? This is also wall-clock, so I'm not sure why things would stop there. This seems to conflate training speed with performance (although also note that the wall clock times being discussed are pretty small -- the rightmost side of the graph is ~14 hours -- hardly a burdensome experiment).
(c) Similarly, for 16x16 cells, SPAIR seems to be improving consistently. Is it being cut off?-Figure 5 -- The caption for the figure things appears to not make sense: GENESIS is listed as having K = HxW+5 components and SPACE has K=5 listed. Neither make sense to me. Are they out of order?
(d) For SPAIR in Table 1, it's not clear whether it's the slow SPAIR that was mentioned previously or the fast one (e.g., the predicted boxes are described as the same quality as SPAIR -- but is it the slow SPAIR or fast one?). I think the paper would benefit from being a bit clearer about this. I get that the parallel decomposition, in some sense, may be necessary to get any results. But I wish the paper were a bit more explicit.


-There are some reasonable results. I realize that there isn't existing ground truth on atari and other games, but why not label a few hundred frames manually?
-It would have been nice to have an ablation of some of the components, including the boundary loss. Unfortunately, there's a complex multi-part system and it's not clear how to break off components apart for reuse elsewhere.

Small stuff that doesn't affect my review:
1) Figure 5 -- the figure text size is tiny and should be fixed.
2) Eqn 3, subscript of the product "i" -> "i=1"
3) Table 1 -- captions on tables go on top
4) Now that the systems work like this, I'd encourage the authors to go and try stuff on more realistic data.
5) I would be a little wary of making a big deal out of the discovery of the Montezuma's Revenge key. I realize this is indeed important, but I don't see why something like the slic superpixel objective, or felzenswalb-huttenlocher wouldn't find it either. I think it's great that there's a setting in terms of network capacity (for fg/bg networks) that yields this result, but this seems to depend heavily on the particular networks used for each of the parts of the method, and not on the general method. Also, it seems largely a function of the fact that they're a small region with a different color.

-----------------------------------------

Post-rebuttal update: I have read the authors' response and I am happy to increase my rating to 6.

**Experience Assessment:**

I have published in this field for several years.

**Review Assessment: Checking Correctness Of Derivations And Theory:**

I assessed the sensibility of the derivations and theory.

**Review Assessment: Checking Correctness Of Experiments:**

I carefully checked the experiments.

**Review Assessment: Thoroughness In Paper Reading:**

I read the paper thoroughly.

---

> ### Author Response · Authors · 2019-11-11
> **Response to Blind Review #2 (1/2)**
>
> “Why does Figure 1 show results from different systems on different images? This makes comparison impossible. Paired samples are always more informative.”
>
> -> We agree that we should present the results from the different methods using the same set of images. In our updated version of the paper, we have made this change.
>
> “(a) It's never listed how K for genesis was picked -- this should presumably be tuned somewhere to optimize performance. The paper mentions in the 4.2 that it was impossible to run the experiments for GENESIS for more than 256 components -- but the GENESIS paper has numbers more like K=9. If there are an overabundance of components, this might explain some of the object splitting observed in the paper.”
>
> -> For qualitative results in Section 4.1, we picked the best K via hyperparameter search for all methods. In this experiment, to focus on the best decomposition quality, we use the knowledge of the number of objects in the scene which is in general not given at test time. For GENESIS specifically, the best values of K we found range from 12 to 65 depending on the environment.
>
> -> For quantitative results in Section 4.2, the goal is to see the performance on decomposition capacity (we’ve updated our paper to denote this as C). The higher decomposition capacity is better. Because we do not know how many objects will be given at test time, we want a single model to have high decomposition capacity rather than having multiple models of different capacities. For example, if we set the capacity to C=256, it means that the model should be able to deal with scenes with the number of objects from 0 to 256. SPACE can flexibly deal with such a broad range of scenes with capacity C=256. In a scene with a single object, SPACE should not split it into multiple objects because of the spatially parallel local detection. In a scene with a large number of objects, say 240, SPACE should reasonably be able to deal with it due to the spatially parallel & local detection mechanism. SPAIR should work similarly but just much more slowly because it is also spatially local but sequential. What if GENESIS or IODINE is given a scene with about 240 objects? To deal with this, it should have capacity like C=256. What if it is then given a scene with a single object? (Again, it is clear that we do not want to use another model trained for e.g., C=10) As claimed in the GENESIS or IODINE paper, in the ideal case, it should learn to only use one component to capture the object while suppressing all other components, instead of splitting a single object into many (256) small parts. This seems pretty difficult due to its sequential nature ignoring spatial locality. Showing this is the goal of the experiment.
>
> “(b) Unless I'm missing something, in Figure 5, for 4x4 and 8x8, it doesn't appear that IODOINE or GENESIS have converged at all. Does the validation MSE just then flatten (or go up) there? This is also wall-clock, so I'm not sure why things would stop there. This seems to conflate training speed with performance (although also note that the wall clock times being discussed are pretty small -- the rightmost side of the graph is ~14 hours -- hardly a burdensome experiment).
> (c) Similarly, for 16x16 cells, SPAIR seems to be improving consistently. Is it being cut off?”
>
> -> Our original intent for these charts was to show that SPACE converges more quickly than the other methods for a given decomposition capacity. However, this is a good point that we can additionally show both the actual time to convergence for each of the methods as well as the MSE to which each method converges. We have updated these charts so they are no longer cut off in our updated version of the paper.
>
> “Figure 5 -- The caption for the figure things appears to not make sense: GENESIS is listed as having K = HxW+5 components and SPACE has K=5 listed. Neither make sense to me. Are they out of order?”
>
> -> We hope this is clarified with our response to (a). We have also updated the paper to clarify these configurations.

---

> > ### Author Response · Authors · 2019-11-11
> > **Response to Blind Review #2 (2/2)**
> >
> > “(d) For SPAIR in Table 1, it's not clear whether it's the slow SPAIR that was mentioned previously or the fast one (e.g., the predicted boxes are described as the same quality as SPAIR -- but is it the slow SPAIR or fast one?). I think the paper would benefit from being a bit clearer about this. I get that the parallel decomposition, in some sense, may be necessary to get any results. But I wish the paper were a bit more explicit.”
> >
> > -> Thank you for pointing this out. These results were run with the faster version of SPAIR. We have updated our paper to make this more explicit. Separately, we’ve also denoted the patch-based version of SPAIR as SPAIR-P to distinguish it from the version of SPAIR that trains on the entire image.
> >
> > “-There are some reasonable results. I realize that there isn't existing ground truth on atari and other games, but why not label a few hundred frames manually?”
> >
> > -> This is a good suggestion. We are in the process of hand-annotating images for Space Invaders (the game for which both SPAIR and SPACE perform well). We will update our paper with the results once we complete our experiment on this data.
> >
> > “-It would have been nice to have an ablation of some of the components, including the boundary loss. Unfortunately, there's a complex multi-part system and it's not clear how to break off components apart for reuse elsewhere.”
> >
> > -> We would like to point out that our experiments varying the background components of SPACE from K=1 to K=5 is actually an ablation study of our background module. Additionally the comparison of SPACE with K=1 with SPAIR can be seen as an ablation study of parallel vs sequential inference of the cell latents. We agree that an ablation study on the boundary loss is also interesting. We have conducted this experiment and included the results in the “Average Precision and Error Rate” section.
> >
> > “I would be a little wary of making a big deal out of the discovery of the Montezuma's Revenge key. I realize this is indeed important, but I don't see why something like the slic superpixel objective, or felzenswalb-huttenlocher wouldn't find it either. I think it's great that there's a setting in terms of network capacity (for fg/bg networks) that yields this result, but this seems to depend heavily on the particular networks used for each of the parts of the method, and not on the general method. Also, it seems largely a function of the fact that they're a small region with a different color.”
> >
> > -> This is a good point. We agree that the key can be detected by using a segmentation method where the objective function is explicitly designed solely for segmentation. We think the detection of the key in SPACE is interesting because the detection emerges using unsupervised neural end-to-end learning where a segmentation algorithm is not explicitly implemented but the goal is to generate (reconstruct). In this setting, from our extensive experiments on SPAIR with background, we found that it is very difficult to detect the key even if it is a small region with a different color. This is because the key always appears in the same position with the same appearance, and thus the background module tends to model it very easily. (This is also observed in Fig 2, SPAIR 16x16 on SpaceInvaders. Here, we see the red obstacle is detected as background in SPAIR while SPACE detects it as a foreground.) Other dynamic objects were detected rather easily by the foreground module. The new knowledge we found from the SPACE experiment is that the background decomposition seems to allow the key to be detected as a foreground object. We think that this is because each of the background components is a weak module. One may ask why not use a weak background module in SPAIR, but since SPAIR only has a single background component, it would not be able to model complex backgrounds. SPACE, on the other hand, would still be able to model complex backgrounds with the cooperation of multiple background components. We nevertheless totally agree that we need more investigation about this interesting phenomenon.

---

> > > ### Author Response · Authors · 2019-11-15
> > > **Blind Review #2 Followup**
> > >
> > > Update on the hand labeling experiment:
> > >
> > > Following the reviewer’s suggestion, we hand-labeled 100 images for SpaceInvaders. However, because of the small size of the objects, the IoU metric we use is very sensitive to small deviations -- just a 2 to 3 pixel difference in height and width between the predicted and ground truth bounding boxes caused the IoU to drop below 0.5. This makes the value of the metric extremely sensitive to the tightness of the boxes as well as the ground truth boxes that we labeled, and thus not a good indicator of the actual bounding box quality.
> > >
> > > As an example, we have uploaded an anonymous image here: https://ibb.co/2MgSzs3. The green boxes are SPACE predicted bounding boxes and the red boxes are ground truth. As can be seen from this image, the predicted bounding boxes are generally larger than the ground truth bounding boxes. Because of the sensitivity of the IoU metric, the AP@0.5 for this image will be low. However, from a qualitative standpoint, one could not objectively say that the ground truth bounding boxes are better than the predicted bounding boxes (with the small bullet being the exception).
> > >
> > > While we did not decide to include these results in the paper at this time, this may be something we can reconsider for a camera-ready version of this paper when we have more time to investigate the results.

---

### Official Review · AnonReviewer3 · 2019-10-26
**Official Blind Review #3**

**Rating:** 3

**Review:**

The paper proposes SPACE: a generative latent variable models for scene decomposition (foreground / background separation and object bounding box prediction). The authors state the following contributions relative to prior work in this space: 1) ability to simultaneously perform foreground/background segmentation and decompose the foreground into distinct object bounding box predictions, 2) a parallel spatial attention mechanism that improves the speed of the architecture relative to the closest prior work (SPAIR), 3) a demonstration through qualitative results that the approach can segment into foreground objects elements that remain static across observations (e.g. the key in Montezuma's Revenge).

The proposed model is evaluated on two sets of datasets: recorded episodes from subsets of the Atari games, and "objects in a 3D room" datasets generated by random placement of colored primitive shapes in a room using MuJoCo.  Qualitative results demonstrate the ability of the proposed model to separate foreground from background in both datasets, as well as predict bounding boxes for foreground objects.  The qualitative results show comparisons against SPAIR, as well as two mixture-based generative models (IODINE and GENESIS), though mostly not for direct comparisons on the same input.  Quantitative results compare the proposed model against the baselines in terms of: gradient step timings, and convergence plots of RMSE of reconstruction against wall clock time, and finally on object bounding box precision and object count error in the 3D room dataset.

The key novelty of the proposed model is that it decomposes the foreground latent variable into a set of latents (one for each detected object), and attends to these in parallel. This leads to improved speed compared to SPAIR, as demonstrated by the gradient step timings. I am convinced that the proposed model is asymptotically faster than SPAIR. However, the fact that the timings are only reported for the gradient step and not more comprehensively for entire training and inference step, is unsatisfying.  I found the qualitative comparisons to be confusing as they were mostly for different input frames, making it hard to have a direct comparison of the quality between the proposed method and baselines.  Moreover, the quantitative results reporting bounding box precision are confusing. Why report precision at exactly IoU = 0.5 and IoU in [0.5, 0.95] instead of the more standard precision at IoU >= 0.5 (and higher threshold values such as 0.95)?  The differences in the reported results seem relatively small and in my opinion, not conclusive given the above unclear points.

Due to the above weaknesses in the evaluation, I am not fully convinced that the claimed contributions are substantiated empirically.  Thus I lean towards rejection.  However, since I am not intimately familiar with the research area, I am open to being convinced by other reviewers and the authors about the conceptual contributions of the model.  As it stands, I don't think this contribution is strong enough to merit acceptance.


**Experience Assessment:**

I have read many papers in this area.

**Review Assessment: Checking Correctness Of Derivations And Theory:**

I assessed the sensibility of the derivations and theory.

**Review Assessment: Checking Correctness Of Experiments:**

I assessed the sensibility of the experiments.

**Review Assessment: Thoroughness In Paper Reading:**

I read the paper at least twice and used my best judgement in assessing the paper.

---

> ### Author Response · Authors · 2019-11-11
> **Response to Blind Review #3**
>
> “However, the fact that the timings are only reported for the gradient step and not more comprehensively for entire training and inference step, is unsatisfying.”
>
> -> With regards to the entire training time, we have shown that our method is clearly faster than the others via the right-most three plots in Figure 5. In these plots, the x-axis is wall-clock time and thus shows the overall convergence rate during training. For the inference step, the first plot in Figure 5 shows that SPACE is also faster than the other methods. To clarify, when we mention the gradient step, we are actually referring to the time for both forward and backward propagation. Therefore, the inference step (which is part of the forward pass) is included in our definition of the gradient step. Also, even without this experiment, from the design of the methods, it should be clear that the comparing baseline methods should be slower than ours. Specifically, our method is parallel except for a small number of background components while other methods are all fully sequential. Thus, it should be clear to see other methods become slower as C (decomposition capacity) increases. The experiments just reaffirm this point to provide actual numbers.  In our updated version of the paper, we made this point more clear.
>
> “I found the qualitative comparisons to be confusing as they were mostly for different input frames, making it hard to have a direct comparison of the quality between the proposed method and baselines.”
>
> -> We agree that we should present the results from the different methods using the same set of images. In our updated version of the paper, we have made this change.
>
> “Moreover, the quantitative results reporting bounding box precision are confusing. Why report precision at exactly IoU = 0.5 and IoU in [0.5, 0.95] instead of the more standard precision at IoU >= 0.5 (and higher threshold values such as 0.95)?”
>
> -> The result in the IoU = 0.5 column uses 0.5 as a threshold rather than an exact value. This is the standard metric used in the Pascal VOC challenge. The [0.5, 0.95] result is the mean average precision over different IoU thresholds from 0.5 to 0.95. More specifically, we take 0.05 increments, so we have average values for the following IoU thresholds: (0.5, 0.55, 0.6, 0.65, 0.7, 0.75, 0.8, 0.85, 0.9, 0.95). This is also the same metric used in the MS COCO object detection challenge. In our updated version of the paper, we make this more clear.
>
> “The differences in the reported results seem relatively small and in my opinion, not conclusive given the above unclear points”
> Our aim is not to show that we can produce better bounding boxes than SPAIR. Rather, we want to show that our method can produce similar quality bounding boxes while 1) having orders of magnitude faster inference and gradient step time, 2) converging more quickly than other methods, 3) scaling to a large number of objects without significant performance degradations, and 4) providing complex background segmentation. In our updated version of the paper, we have emphasized the above in the quantitative evaluation section to make this more clear.

---

### Official Review · AnonReviewer4 · 2019-10-31
**Official Blind Review #4**

**Rating:** 6

**Review:**



In this paper, the authors propose a generative latent variable model, which is named as SPACE, for unsupervised scene decomposition. The proposed model is built on a hierarchical mixture model: one component for generating foreground and the other one for generating the background, while the model for generating background is also a mixture model. The model is trained by standard ELBO with Gumbel-Softmax relaxation of the binary latent variable. To avoid the bounding box separation, the authors propose the boundary loss, which will be combined with the ELBO for training. The authors evaluated the proposed on 3D-room dataset and Atari.


There are several issues need to be addressed:


1, The organization of the paper should be improved. For example, the introduction to the generative model is too succinct: the spatial attention model did not be introduced in main text. Why this model is call 'spatial attention' is not clear to me. The boundary loss seems an important component, however, it is never explicitly presented.

2, The parallel inference is due the mean-field approximation, in which the posterior is approximated with factorized model, therefore, the flexibility is restricted. This is a trade-off between flexibility and parallel inference. The drawback of such parametrization should be explicitly discussed. I was wondering is there any negative effect of such the approximated posterior with fully factorized model comparing to the SPAIR?

3, The empirical evaluation is not convincing. The quality illustration in Fig.1, 2 and 3 uses different examples for different methods. This cannot demonstrate the advantages of the proposed model. The quantitative evaluation only shows one baseline, SPAIR, in Table 1, and other baselines (IODINE and GENESIS) are missing. With such empirical results, the performances of the proposed method are not convincing.

In sum, I think this paper is not ready to be published.

====================================================================

I have read the authors' reply and the updated version. I will raise my score to 6.

Although the mean-field inference is standard, the model in the paper looks still interesting and the performances are promising.

I expect the boundary loss should be specified formally in the final version.

**Experience Assessment:**

I do not know much about this area.

**Review Assessment: Checking Correctness Of Derivations And Theory:**

I assessed the sensibility of the derivations and theory.

**Review Assessment: Checking Correctness Of Experiments:**

I assessed the sensibility of the experiments.

**Review Assessment: Thoroughness In Paper Reading:**

I read the paper at least twice and used my best judgement in assessing the paper.

---

> ### Author Response · Authors · 2019-11-11
> **Response to Blind Review #4**
>
> “1, The organization of the paper should be improved. For example, the introduction to the generative model is too succinct: the spatial attention model did not be introduced in main text. Why this model is call 'spatial attention' is not clear to me.”
>
> -> We agree that we can be more clear about our generative model and how spatial attention is used. In our updated version of the paper, we clarified the description and included a diagram (see Figure 1) that better illustrates our model and depicts how spatial attention is used. We also provide a more thorough discussion of parallel spatial attention in the “Parallel Inference of Cell Latents” section.
>
> “The boundary loss seems an important component, however, it is never explicitly presented.”
>
> -> We present the boundary loss in the “Preventing Box-Splitting” subsection and provide implementation details in Appendix C. We’ve also included an ablation study that removes boundary loss in our Average Precision and Error Rate experiments.
>
> “2, The parallel inference is due the mean-field approximation, in which the posterior is approximated with factorized model, therefore, the flexibility is restricted. This is a trade-off between flexibility and parallel inference. The drawback of such parametrization should be explicitly discussed. I was wondering is there any negative effect of such the approximated posterior with fully factorized model comparing to the SPAIR?”
>
> -> We agree that the independence assumption for mean-field may provide a poor approximation in many cases. In our problem, one potential negative effect of this assumption is that it may result in duplicate detections due to objects not referring to each other. This independence assumption, however, is not always a poor approximation. It is actually a good choice if the underlying system actually shows weak dependency between the factors, which is actually the case in our problem. We observe that -- contrary to what the SPAIR authors conclude by saying that sequential processing is crucial for performance -- the independence assumption should not affect the performance much. This is because we observe that (1) due to the bottom-up encoding conditioning on the input image, each object latent should know what’s happening around its nearby area without communicating with each other, and that (2) in (physical) spatial space, two objects cannot exist at the same position. Thus, the relation and interference from other objects should not be severe. Based on this reasoning, questioning what the SPAIR authors concluded, we implement the parallelization and show empirically that our insight and reasoning are actually correct by showing comparable performance to SPAIR. Importantly, via a recent personal communication, the SPAIR authors also confirmed that they also recently realized that the independence assumption is correct (and thus parallelizable) even if they didn’t know it when they had published SPAIR.
>
> We believe that this can also be considered a contribution because it corrects the previous state-of-the-art knowledge that was against the possibility of parallel processing.
>
>
> “3, The empirical evaluation is not convincing. The quality illustration in Fig.1, 2 and 3 uses different examples for different methods. This cannot demonstrate the advantages of the proposed model.”
>
> -> We agree that we should present the results from the different methods using the same set of images. In our updated version of the paper, we have made this change.
>
> “The quantitative evaluation only shows one baseline, SPAIR, in Table 1, and other baselines (IODINE and GENESIS) are missing. With such empirical results, the performances of the proposed method are not convincing.”
>
> -> For our quantitative evaluation, we compare gradient step latency and time to convergence for all methods. However, since IODINE and GENESIS do not produce bounding boxes, we cannot compare Average Precision and Error Rate for those methods. We would also like to emphasize that our aim is not to necessarily produce better bounding boxes than SPAIR. Rather, we want to show that our method can produce similar quality bounding boxes while 1) having orders of magnitude faster inference and gradient step time, 2) converging quicker than other methods, 3) scaling to a large number of objects without significant performance degradation, and 4) providing complex background segmentation. In our updated version of the paper, we have emphasized the above in the quantitative evaluation section to make this more clear.

---

### Official Review · AnonReviewer1 · 2019-11-08
**Official Blind Review #1**

**Rating:** 6

**Review:**

This paper studies the problem of unsupervised scene decomposition with a foreground-background probabilistic modeling framework. Building upon the idea from the previous work on probabilistic scene decomposition [Crawford & Pineau 2019], this paper further decomposes the scene background into a sequence of background segments. In addition, with the proposed framework, scene foreground-background interactions are decoupled into foreground objects and background segments using chain rules. Experimental evaluations have been conducted on several synthetic datasets including the Atari environments and 3D-Rooms. Results demonstrate that the proposed method is superior to the existing baseline methods in both decomposing objects and background segments.

Overall, this paper studies an interesting problem in deep representation learning applied to scene decomposition. Experimental results demonstrated incremental improvements over the baseline method [Crawford & Pineau 2019] in terms of object detection. However, reviewer has a few questions regarding the intuition behind the foreground-background formulation and the generalization ability to unseen combinations or noisy inputs.

== Qualitative results & generalization ==
The qualitative improvements over the baseline method [Crawford & Pineau 2019] seem not very impressive (Figure 1: only works a bit better with cluttered scenes). First, how does the proposed method perform in real world datasets (Outdoor: KITTI, CItyscape; Indoor: ADE20K, MS-COCO)? Second, the generalization to unseen scenarios are mentioned in the introduction but not really carefully studied or evaluated in the experiments. For example, one experiment would be to train the framework on the current 3D-Rooms dataset but then test on new environments (e.g., other room layout) or new objects (e.g. other shapes such as shapenet objects).


== Application beyond object detection ==
Equation (4) does not seem to be natural in practice: basically, the background latents depends on the foreground object latents. Alternatively, you can assume them to be independent with each other. It’s better to clarify this point in the rebuttal. As this is a generative model, reviewer would like to know the applicability to other tasks such as pure generation, denoising and inpainting. For example, how does the pre-trained model perform with noisy input (e.g., white noise added to the image)? Also, what’s the pure generation results following the chain rules given by Equation (1), (3) & (4).



**Experience Assessment:**

I have published in this field for several years.

**Review Assessment: Checking Correctness Of Derivations And Theory:**

I assessed the sensibility of the derivations and theory.

**Review Assessment: Checking Correctness Of Experiments:**

I carefully checked the experiments.

**Review Assessment: Thoroughness In Paper Reading:**

I read the paper thoroughly.

---

> ### Author Response · Authors · 2019-11-15
> **Response to Blind Review #1 (1/2)**
>
> “The qualitative improvements over the baseline method [Crawford & Pineau 2019] seem not very impressive (Figure 1: only works a bit better with cluttered scenes).”
>
> ->  This is true. However, as noted in responses to other reviewers, we would like to emphasize that our aim is not necessarily to produce better bounding boxes than SPAIR. Rather, we want to show that our method can produce similar quality bounding boxes while achieving the following important properties: 1) having orders of magnitude faster inference and gradient step time, 2) learning more quickly than other methods, 3) scaling to a large number of objects without significant performance degradation, and 4) providing complex background segmentation. In our updated version of the paper, we have emphasized the above in the quantitative evaluation section to make this more clear.
>
> “How does the proposed method perform in real world datasets (Outdoor: KITTI, CItyscape; Indoor: ADE20K, MS-COCO)?”
>
> -> This is a good question. First, we want to note that compared to previous models like AIR, SPAIR, IODINE, and GENESIS which use relatively simple datasets (few objects with a simple background), our dataset and task are far more complex, consisting of many objects (>20 in 3D-Room Large) with complex and dynamic backgrounds (Atari). This level of tasks has not been tested before in the related works. Thus, we believe our task is a fairly challenging one from the perspective of unsupervised generative representation learning of object representations. However, despite the above improvement over previous models, we believe that the proposed model and more generally unsupervised approaches to decomposed object representation learning, still require significant innovations to make it applicable to “in-the-wild datasets.” We believe that achieving this ability would require more unsupervised or self-supervised learning signals like interacting with objects and temporal observations.
>
> “Second, the generalization to unseen scenarios are mentioned in the introduction but not really carefully studied or evaluated in the experiments. For example, one experiment would be to train the framework on the current 3D-Rooms dataset but then test on new environments (e.g., other room layout) or new objects (e.g. other shapes such as shapenet objects).”
>
> This is an interesting question. Again, we want to emphasize that SPACE is an unsupervised generative model, which means that it is actually modeling the distribution that generates the dataset. Thus, we only expect generalization within this distribution. So, as we demonstrate in our experiments, it can generalize to unseen 3D Room scenes where the number, color and shape combinations, and placements of the objects are never seen in training, but it is difficult, in theory, to generalize to unseen scenes that are completely different from the ones we saw in the training set.
>
> “Equation (4) does not seem to be natural in practice: basically, the background latents depends on the foreground object latents. Alternatively, you can assume them to be independent with each other. It’s better to clarify this point in the rebuttal.”
>
> -> We agree that, although our experimental results show that it works quite well in practice, this may be a bit unnatural. Also, we agree that assuming them to be independent, or conditioning foreground latents on background latents are also reasonable choices, especially for generation. In the paper, however, our main focus is learning object representation, not generation. While the proposed modeling also has the potential to generate, like other works (AIR, SPAIR), we do not optimize the model toward that direction to focus on the main contribution. (as discussed in more detail below).

---

> > ### Author Response · Authors · 2019-11-15
> > **Response to Blind Review #1 (2/2)**
> >
> > “As this is a generative model, reviewer would like to know the applicability to other tasks such as pure generation, denoising and inpainting. For example, how does the pre-trained model perform with noisy input (e.g., white noise added to the image)? Also, what’s the pure generation results following the chain rules given by Equation (1), (3) & (4)..”
> >
> > First, we want to note that the focus of this work is not generation. Rather, similar to other generative models (AIR and SPAIR) for object representation learning, we focus on inference, in which we decompose the scene into meaningful components defined in the generation process and produce a good representation for each of them. While it is possible for our model to have unconditional generation by sampling from the priors, due to the independence assumption in our model, each object would be generated and placed independently of other objects in the scene, resulting in unrealistic generation. For example, if we generate in Atari scenes, an object can be placed anywhere, and hence, would not be coherent with the actual game. For the same reason, we did not investigate applications like denoising or inpainting. Similarly, the related works AIR, SPAIR, and IODINE do not provide generation results in their papers. That being said, we believe investigating structured scene models whose main purpose is generation would be very interesting in future research.

---

### Author Response · Authors · 2019-11-11
**For All Reviewers**

We want to thank all the reviewers for taking the time to read our paper and provide insightful feedback. We have prepared responses for the first three reviewers (Reviewers #2, #3 and #4) and we will address Reviewer #1’s feedback shortly. We have uploaded a new version of the paper, which addresses the questions and concerns that were raised. Specifically, we have updated the following:

1) Updated our qualitative experiments to use the same set of images across all different methods.
2) Included a diagram (Figure 1), that better illustrates our model and depicts how spatial attention is used.
3) Clarified a few points in the Quantitative Comparison section with regards to decomposition capacity (C) and how we chose C for the baselines. Also further emphasized that our goal is not to necessarily produce better bounding boxes than SPAIR, but rather to show that we can still produce similar quality bounding boxes while taking advantage of a parallel architecture and providing complex background decomposition.
4) Updated the convergence charts so no methods are cut off.
5) Updated our Average Precision and Error Rate experiments to include an ablation study of boundary loss. For table 1, we also further optimized the hyperparameters for SPACE and SPAIR. Specifically, we found that the scale prior (which controls the tightness of the boxes) and sigma (for computing likelihood) can significantly affect the results, so we tuned both models to make both average precision and error rates as good as possible.
6) Minor edits for typos, formatting, and clarity, including those pointed out by the reviewers. Additionally, we moved a few of the qualitative images to the appendix in the interest of space.

We have also created a website with additional qualitative examples and video of SPACE: https://sites.google.com/view/space-project-page/home

We will respond to each reviewer’s points in detail in the comments below. We believe we have addressed each reviewer’s concerns and look forward to hearing feedback about the updated version of our paper. We hope the reviewers can take our responses and revisions into consideration when evaluating our final score.

---

### Decision · Program_Chairs · 2019-12-19

**Decision:**

Accept (Poster)

**Comment:**

The paper makes a reasonable contribution to generative modeling for unsupervised scene decomposition.  The revision and rebuttal addressed the primary criticisms concerning the qualitative comparison and clarity, which caused some of the reviewers to increase their rating.  I think the authors have adequately addressed the reviewer concerns.  The final version of the paper should still strive to improve clarity, and strengthen the evaluation and ablation studies.